# Focus Review on Nanomaterial-Based Electrochemical Sensing of Glucose for Health Applications

**DOI:** 10.3390/nano13121883

**Published:** 2023-06-19

**Authors:** Domenica Tonelli, Isacco Gualandi, Erika Scavetta, Federica Mariani

**Affiliations:** Department of Industrial Chemistry “Toso Montanari”, University of Bologna, Viale del Risorgimento 4, 40136 Bologna, Italy; domenica.tonelli@unibo.it (D.T.); erika.scavetta2@unibo.it (E.S.); federica.mariani8@unibo.it (F.M.)

**Keywords:** glucose, nanomaterials, electrochemical sensors, diabetes

## Abstract

Diabetes management can be considered the first paradigm of modern personalized medicine. An overview of the most relevant advancements in glucose sensing achieved in the last 5 years is presented. In particular, devices exploiting both consolidated and innovative electrochemical sensing strategies, based on nanomaterials, have been described, taking into account their performances, advantages and limitations, when applied for the glucose analysis in blood and serum samples, urine, as well as in less conventional biological fluids. The routine measurement is still largely based on the finger-pricking method, which is usually considered unpleasant. In alternative, glucose continuous monitoring relies on electrochemical sensing in the interstitial fluid, using implanted electrodes. Due to the invasive nature of such devices, further investigations have been carried out in order to develop less invasive sensors that can operate in sweat, tears or wound exudates. Thanks to their unique features, nanomaterials have been successfully applied for the development of both enzymatic and non-enzymatic glucose sensors, which are compliant with the specific needs of the most advanced applications, such as flexible and deformable systems capable of conforming to skin or eyes, in order to produce reliable medical devices operating at the point of care.

## 1. Introduction

Glucose sensing is of paramount importance for a variety of healthcare related applications, including the early detection and monitoring of chronic diseases, among which diabetes is the primarily targeted one as it stands nowadays as one of the major causes of death and disability worldwide. With 463 million patients estimated in 2019, projections show that such a number is expected to increase by 10.2% in 2030 [1]. Moreover, the management of all the related health complications requires tight personal control of blood glucose daily, thus making glucose the most commonly tested analyte [2] and originating a huge market size for glucose biosensing point-of-care devices. Historically, the development of new biosensing strategies, along with the evolution and commercialization of glucose biosensors, has been strongly driven by the global need to cope with one of the most challenging societal and healthcare emergencies of all times. Moreover, the maintenance of optimal mental and physical performance, together with healthy nutritional habits [3], and wound healing monitoring [4] are other important applications for which glucose monitoring plays a key role.

The design of the first glucose biosensor dates back to 1962 with the pioneering work of Clark and Lyons [5], when a thin layer of concentrated glucose oxidase (GOx) was trapped in between two dialysis membranes in contact with a pH sensitive glass or a pO_2_ sensing electrode. The sensing mechanisms exploited either the pH variation or the O_2_ consumption, respectively, following the enzymatic oxidation of glucose to gluconic acid. As a matter of fact, the electrochemical transduction of a hardly oxidizable substrate such as glucose and the use of enzymes to extend the specificity of electrode systems were demonstrated. This paved the way for the subsequent development of the three well-known generations of glucose biosensors, and greatly implemented the accuracy and precision of the glucose sensing systems for diabetes care, available at that time. Non-enzymatic urine glucose tests were, in fact, already available to the market since the beginning of the 20th century and the first enzymatic test strip was commercialized in mid-60s, both of them based on a colorimetric method providing semi-quantitative information only [6].

The first generation of biosensors was based on the amperometric detection of the H_2_O_2_ produced during the enzymatic oxidation of glucose, occurring at the GOx layer encapsulated in between two membranes that served as physical barriers against larger interfering molecules. The amperometric detection occurred at a Pt electrode at relatively high applied voltages [7]. The push towards the mass production of cheaper and smaller sensing devices, together with the drawbacks associated with the use of high potentials that facilitate crosstalk in whole blood, led to the development of the second generation of glucose biosensors. Their structure is characterized by the presence of small redox active molecules, e.g., redox mediators, which replace the natural oxygen co-substrate and act as non-physiological electron acceptors capable of diffusing across the protein shell towards the active site of the enzyme to re-oxidize it, and then diffuse back shuttling electrons between GOx and the electrode surface. The first and most popular example was the ferrocene-based glucose sensor reported by Cass et al. in 1984 [8]. The major advancement provided by the use of redox mediators was the possibility to detect glucose at lower applied potentials, thus significantly decreasing the interference of easily oxidizable compounds present in real matrices, and in a wider linear range, especially at high concentrations where stoichiometric limitation of oxygen can occur. First examples of further improvements in promoting charge transfer phenomena were achieved through enzyme wiring with redox polymers [9,10], chemical modification of the enzyme [11,12] and the use of nanomaterials [13]. However, the toxicity of most mediators and the thrilling opportunity to realize a direct electron transfer (DET) in the absence of a mediator pushed researchers’ efforts towards the design of the third generation of biosensors. On one hand, further simplification of the sensing system is obtained by removing another reagent from the reaction sequence. On the other hand, superior selectivity can be, in principle, achieved due to the possibility to operate in a potential window closer to the redox potential of the enzyme itself. For this, an especially short distance must be kept between the recognition element and the transducer and most strategies rely on organic conducting materials based on charge-transfer complexes [14,15,16,17]. Although DET has been invoked multiple times in the literature, only a few papers actually provide proof of the mediator-free detection [2], and no third-generation glucose biosensor has been commercialized so far.

Although the use of an enzymatic recognition element is evidently beneficial in terms of selectivity, it also represents an additional cost, and its activity is strongly pH and temperature dependent. With an alternative approach, non-enzymatic glucose sensors have been realized exploiting electrocatalytic phenomena occurring at the functionalized electrode surface, and the two main models proposed to explain enzyme-free detection mechanisms are well summarized in a recent review by Sehit and Altintas [18]. Beyond the thoroughly studied enzymatic and electrocatalytical approaches, different transduction strategies have been explored, because glucose sensing represents a benchmark in the field. Aptamers [19,20,21] and molecularly imprinted polymers [22] are examples of recognizing elements in glucose sensors.

The availability of nanomaterials with controllable and defined size and shape, as well as reproducible physicochemical properties, has profoundly impacted the field of electroanalytical sensing thanks to the easy engineering of chemically modified electrodes. Nanoparticles, nanostructured thin films, inorganic-organic hybrids and composites, nano-alloys, self-assemblies, graphene and carbon nanotubes are some representative classes of nanomaterials exhibiting a uniquely high surface area, enhanced exposure to binding sites and electron transfer properties, together with easy functionalization, which lead to the fine-tuning of (bio)sensing architectures and assist recognition phenomena occurring at the nanoscale level [18,23]. Moreover, nanomaterials can be designed to obtain antifouling proprieties [24,25] and, thus, to increase the lifetime of the sensors when they operate in harsh conditions for long time. Wearable devices should monitor glucose in a biofluid containing a lot of biomolecules that can interact with the sensing surface leading to the creation of films that can vary the response. Due to their unique features, nanomaterials have been successfully applied to both enzymatic and non-enzymatic glucose sensors and show great potential towards the development of minimally or non-invasive diagnostic devices [26], enabling continuous glucose monitoring (CGM). The routine management of diabetes is, in fact, still largely based on the finger-pricking method, which is usually considered unpleasant, can be the source of infections, and does not warn in case of hypoglycaemia events. In alternative, CGM systems mostly rely on the electrochemical sensing of glucose in the interstitial fluid (ISF) by a small transcutaneous needle-based electrode, placed under the skin, and real-time glycemic data are transmitted to a portable receiver. Since 1999, when the first CGM system (MiniMed) received the approval of the U.S. Food and Drug Administration, the accuracy of commercial CGM systems has rapidly been approaching the self-monitoring of blood glucose golden standard. Nevertheless, glucose monitoring in naturally secreted biofluids that can be non-invasively sampled, such as sweat, is currently one of the most intriguing research goals in the field of wearable sensors [27].

The aim of this review is to give a comprehensive overview of the most relevant advancements in glucose sensing achieved in the last 5 years. In particular, both consolidated and innovative electrochemical sensing strategies, based on nanomaterials, will be thoroughly discussed, analyzing performances, advantages and limitations and targeting the definition of the major analytical challenges in glucose detection, in blood and serum samples, as well as in less conventional biological fluids.

## 2. Blood and Serum Samples

### 2.1. Enzymatic Electrochemical Sensors

At present, commercial sensors for diabetes, generally implying a needle pinching to release blood as stated in the Introduction, are mainly based on glucose oxidase, and less frequently on glucose dehydrogenase (GDH). The former enzyme is highly selective towards glucose molecules and more stable than other enzymes, but its activity is strongly dependent on the pH, temperature and dissolved oxygen concentration. In particular, it rapidly loses its activity below pH 2 or above pH 8 and can be damaged at temperatures over 40 °C. On the contrary, the different types of GDH enzymes, each one working with various co-factors, e.g., pyrroloquinoline, nicotine adenine dinucleotide (NAD) and flavin adenine dinucleotide (FAD), catalyze a reaction independent of oxygen [28], but GDH-based sensors can give biased glucose measurements, due to their reactivity towards sugars other than glucose, such as maltose and xylose [29].

Gox-based electrodes convert glucose to gluconic acid and hydrogen peroxide, and, as already stated in the introduction, the commercial ones are always first-generation biosensors which work by exploiting the H_2_O_2_ amperometric detection. The oxidation of hydrogen peroxide may lead to signal interference from other species (e.g., ascorbic acid (AA), uric acid (UA), etc.) susceptible to electro-oxidation, thus compromising the enzymatic selectivity. On the other hand, the quantification of glucose on the basis of oxygen consumption, oxygen being the co-substrate of the enzymatic reaction, depends on the stability of the dissolved O_2_ in the analyzed sample; both these events may lead to inaccurate measurements [30].

In addition, the GOx-based sensors are limited by enzyme immobilization techniques since enzymes are easily inactivated during the immobilization process; therefore, good stability and reproducibility are difficult to achieve, and researchers have been focusing their efforts on this step of biosensor fabrication. Among the many immobilization methods, cross-linking is still used, especially employing low-cost reagents, such as glutaraldeyde (GA) and bovine serum albumin (BSA). Furthermore, in the last few years various biosensors based on graphene (G) have been developed which take advantage of its high specific surface area, in addition to its optimal electrical conductivity, to favor the enzyme immobilization thanks to G adsorption ability [31]. In fact, graphene can be considered not only an outstanding platform to support different biological molecules and nanomaterials, such as metal or metal oxides nanoparticles (NPs), but also an efficient interface between GOx or GDH and the underlying conductive support so as to develop third-generation electrochemical biosensors. In 2021, a minireview was published reporting the results achieved by graphene-based nano(bio)sensors for the determination of glucose in human body fluids [32]. Obviously, carbon nanotubes (CNTs) can also be employed for the development of enzymatic biosensors, exploiting more or less the same peculiar properties as exhibited by graphene. A paper by Jeon et al. described a nano-biocomposite composed of multi-walled carbon nanotubes (MWCTs), chitosan (CS) and GDH which was absorbed onto screen-printed carbon electrodes (SPCEs) for continuous glucose monitoring [28]. Due to the hydroxyl and amine functional groups possessed by chitosan, strong hydrogen bonds are formed with the enzyme, and these interactions generate a sort of core–shell structure, since the CNTs are completely wrapped by CS and the GDH molecules, which allows for the DET from the enzyme to the electrode surface, i.e., without a mediator. In such a way, the GDH/CS/MWCNTs/SPCEs exhibited catalytic oxidation currents at −0.422 V vs. Ag/AgCl that increased with glucose concentration within the range 0–5.5 mM at pH = 7.00, with a limit of detection (LoD) of 0.0156 mM (S/N = 3) and were not affected by interfering compounds such as AA, UA and dopamine (DA) or oxygen. Finally, the immobilization system chosen by the authors assured the stability of the sensor which was demonstrated over 10 days.

Furthermore, metal nanoparticles have also been employed to modify enzyme-functionalized electrodes, and platinum nanoparticles (PtNPs) have proved to be efficient catalysts in electrochemical reactions due to their high electrical conductivity and adsorption ability. A recent work reported the modification of a glassy carbon electrode (GCE) with a suspension of Pt NPs, G and Nafion to support GOx, which was later immobilized by the above cited classical cross-linking method [33]. The hydrophobic perfluorocarbon chains of Nafion aimed to favor the G dispersion, and the sulfonic groups prevented the stacking of G layers; at the same time, Nafion polyanionic nature increased the selectivity of the glucose biosensor towards the tested interferents: AA, UA and DA. The determination of glucose was carried out in synthetic blood samples at +0.60 V vs. Ag/AgCl/3.0 M KCl in phosphate buffer (pH = 8.0) in order to oxidize H_2_O_2_, and the authors, aware of the high cost of GOx, performed a study on the re-usability of the sensor for one month, after which an enzyme retention activity greater than 75% was demonstrated.

Another paper has reported a glucose biosensor exploiting a GCE modified with a nanolayer of electrodeposited Au, on which a nanocomposite made of silver nanoflowers (AgNFs) decorated with Pt NPs and covered with BSA was casted [34]. The BSA acted as a platform to immobilize GOx through the -NH_2_ functional groups cross-linked with GA, and to anchor the nanocomposite to the Au nanolayer thanks to the reaction with the -SH groups. The petal structures of the AgNFs with wide surface areas led to a high electron conductivity. Pt NPs enhanced the electrocatalytic ability toward the reduction of H_2_O_2_. The morphological characterization obtained by scanning electron microscopy (SEM), field emission scanning electron microscopy (FESEM), transmission electron microscopy (TEM) and the elemental energy dispersive X-ray spectroscopy (EDS) maps are shown in Figure 1.

The calibration curve of AgNFs-Pt@BSA/GA/GOx sensors for glucose was obtained in O_2_ saturated phosphate-buffered saline (PBS) solution (pH = 7.4) and displayed a linear relationship in a wide range (from 1 to 14 mM), with a detection limit of 0.3 mM. No significant change in the recorded current was observed when adding glycine, L-glutamic acid, L-valine, L-isoleucine, L-asparagine and L-leucine, as interferents, to a PBS solution containing glucose at the same concentration (5 mM), thus confirming the optimal selectivity of the biosensor, in addition to good fabrication reproducibility and a stability of 10 days. The AgNFs-Pt@BSA/GA/GOx sensor was applied for the determination of glucose levels in human serum samples, displaying a high accuracy [34].

Highly fluorescent Ag nanoclusters decorated with deoxyribonucleic acid (DNA) flowers have been exploited to produce a dual-mode glucose biosensor. The hydrogen peroxide generated by GOx enzymatic reaction etches Ag nanomaterial to produce Ag^+^ that is quantified by a decrease in the fluorescence signal in a signal-off mode. Moreover, the DNA probes are functionalized with ferrocene. Based on a coordination interaction of cytosine-Ag^+^-cytosine, the liberated Ag^+^ also induces a conformation switch of ferrocene-labeled DNA which places the redox probe close to electrode surface allowing the ferrocene oxidation that generates the electrochemical signal. The transduction principle has been validated by glucose detection in human serum with good accuracy and reliability [35].

Additionally, nanoporous materials such as nanoporous gold (nPG) or porous silicon (PS) layer with their peculiar continuous nanostructure and catalytic properties are attractive for the production of electrochemical sensing platforms. nPG has recently received great attention because of its high surface-to-volume ratio which favors the interactions with analytes, high electrical conductivity, selectivity, and antifouling capacity [36].

A very recent paper describes highly sensitive and selective lactate and glucose microbiosensors for the in vivo determination of both analytes in rat brain, which were also applied to the simultaneous determination of glucose and lactate in blood serum samples. The microbiosensor was based on carbon fiber microelectrodes (CFM) modified with a film of nPG, on which Pt NPs were electrodeposited to increase the electrocatalytic efficiency for hydrogen peroxide determination. For the glucose measurement the platform was modified with GOx and a good selectivity towards AA, DA, and UA was obtained by using a permselective layer of electropolymerized *m*-phenylenediamine that prevented large molecules from reaching the electrode surface, followed by the deposition of a Nafion film which excluded the negatively charged molecules. The PtNP/nPG/CFM showed a high sensitivity to H_2_O_2_ (5960 μA mM^−1^ cm^−2^) at 0.36 V vs. Ag/AgCl, with a linear range from 0.2 to 200 μM and a LoD of 10 nM, which led to a glucose linear response in the range 0–1 mM and a LoD of 13 ± 2 μM for the GOx/PtNP/nPG/CFM biosensor [37].

In addition, the peculiar features of nPG were exploited to fabricate low-cost portable glucose biosensors based on screen-printed electrodes (SPE). Upon immobilization of a NAD^+^-dependent glucose dehydrogenase on the nPG/SPE surface, glucose could be determined in human serum samples. The sensors were based on the NADH electro-oxidation produced after the addition of glucose, which occurred with optimal analytical performance within a wide pH range, even if the measurement was performed in PBS at pH 8.8. The procedure employed a microliter sample volume (50 µL), the linearity was demonstrated in the range from 100 μM to 3.0 mM, and the LoD resulted 15 µM. The GDH/nPG/SPE biosensing platform proposed in that study can be considered a promising candidate for the clinical blood test, urine analysis, and mass production potential. The accuracy of the quantification was verified by comparing the results with those obtained with an automatic biochemical analyzer. The analytical performance of the sensor was compared by the authors with those of some GDH-based electrochemical devices recently published, concluding that the GDH/nPG/SPE behavior was even better [38].

Porous Si has a sponge-like structure made of pores and channels surrounded by a network of crystalline Si nanowires. It can be readily produced by electrochemical etching techniques and is becoming a very promising material for sensing applications due to its high surface-to-volume ratio (~500 m^2^ cm^−3^), biocompatibility, low cost and ease of surface functionalization. These peculiar properties make PS suitable for immobilizing a high amount of enzymes in a stable way by physisorption, thanks to its rough surface. The selective determination of glucose has been obtained by surface functionalization of nano PS layers with GOx in conditions of hypo and hyper glycemia (0.12–18 mM glucose concentration) [39]. The basic principle of the resulting biosensor is the change of electrical conductivity in the presence of glucose due to the electron flow generated by the oxidation of hydrogen peroxide at pH = 7.0, which is produced by the enzymatic reaction. Different charge transport mechanisms operate with PS depending on the direct current applied voltage. The scheme of the biosensor is shown in Figure 2a. The measurement was carried out by spraying onto the sensor surface 10 μL of a glucose solution and then drying at room temperature for 2 min. The sensor could be used more than once, and the glucose level could be determined in human saliva, urine and blood. In order to reset the platform for multiple uses, the adsorbed glucose molecules were removed by sodium acetate buffer and tween-20 solution followed by drying. The response time was very short (~170 ms) and the repeatability very good.

However, the above-described glucose sensors require a power supply providing the electricity needed for making the devices operative. Torrinha et al. published the first study that employed pencil graphite electrodes (PGEs) in the development of self-powered biosensors [40]. They developed two bio-electrodes based on a PGE modified with MWCNTs and exploited the bifunctional crosslinker 1-pyrenebutanoic acid succinimidyl ester (PBSE) as tethering agent, enabling the establishment of DET between PGEs and both enzymes proposed for making the biofuel cell operative. In particular, they used the enzyme quinoprotein glucose dehydrogenase (PQQGDH) for the development of the bio-anode, and Bilirubin oxidase (BOx) for the bio-cathode. The latter is a multicopper oxidase enzyme often chosen for the four-electron reduction of oxygen, due to its excellent activity at neutral pH. The two electrodes were conjugated into a biofuel cell which was tested to prove a proper functioning without the connection to an external potentiostat. Furthermore, the PGE/MWCNT/PBSE/PQQGDH cell was also characterized as a biosensor for glucose in 0.1 M phosphate-buffered solution, pH 7, displaying a linear range up to 0.8 mM, a LoD of 4.0 ± 2.2 μM and a stable response for 12 days. The main conclusion of the work is that PGE exploration for microenergy production was successful, and self-powered biosensors might have a great impact in the near future for the marketing of portable devices. The scheme of the self-powered biosensor based on PGE transducer is shown in Figure 2b.

A further evolution of the potential of self-powered biosensors for glucose quantification has been published by Lee et al. [41]. The authors have proposed a flexible, disposable, and portable glucose biosensor where a FAD-dependent GDH enzyme was the biocatalyst for the glucose oxidation whereas the cathode exploited the reduction of Prussian blue (PB) to Prussian white. The electrode material was indium tin oxide (ITO) and it was patterned to produce three bars for the electrodeposition of PB, whereas the fourth bar worked as a bioanode and was fabricated by applying a suspension of MWCNTs on the ITO surface to favor the immobilization of GDH. Moreover, tetrathiafulvalene was used as a redox mediator and a Nafion solution as a binder to immobilize the above three materials to the conductive support. The device, consisting of the patterned ITO, was vertically contacted with a lateral flow membrane, and the entire system was coated with a polyethylene (PET) film, in which a hole was made under the bioanode to allow the glucose solution and electrolyte to enter the lateral flow membrane. The principle of the glucose biosensor (Figure 2c) is that the blue color of PB is turned transparent when the electrons coming from the glucose oxidation arrive to the cathode. The self-powered biosensor was sensitive only to glucose even in the presence of many interfering compounds and displayed a linearity from 0.5 to 10 mM glucose concentration (R^2^ = 0.999). When 3, 5 and 10 mM glucose solutions corresponding to low, normal, and high blood glucose level are injected into the device, each level is easily recognized by the naked eye thanks to the number of PB bars that fade [41].

### 2.2. Non-Enzymatic Electrochemical Sensors

Owing to some drawbacks of the enzyme-based sensors for glucose in practical use, especially related to the cost, the limited storage conditions and the difficulty in maintaining the stability of the enzymes, a large number of studies on catalytic materials for enzyme-free glucose devices have emerged in the last 5 years [42].

In non-enzyme electrochemical sensors, the detection performance mainly depends on the electrocatalytic activity of the electrode materials (Figure 3a), so the research has focused on the design of materials used to modify the conductive support and acting as excellent mediators for glucose oxidation.

To this aim, different kinds of materials have been proposed to develop non-enzymatic glucose sensors including metals (Pt, Au, Cu, etc.), alloys (Pt–Pd, Pt–Au, Cu–Co, etc.), metal oxides (CuO, NiO, Co_3_O_4_, etc.), carbon nanomaterials, metal organic frameworks (MOF) and hybrid materials deriving from a combination of the above-mentioned ones.

#### 2.2.1. Metal Oxides Based Sensors

To reduce the cost of sensors, it is necessary to develop highly sensitive, reliable, but inexpensive materials. Copper and its oxides/hydroxides are the main materials employed for non-enzymatic glucose sensors due to their high catalytic activity, and in recent years, a number of papers have been published describing the successful development of glucose sensors based on Cu, CuO, Cu_2_O, Cu(OH)_2_, Cu-MOF nanomaterials.

A device exploiting a composite material consisting of nano-coral arrays of CuO grown on nanoporous Cu (NCA/NPC) was fabricated which displayed a high sensitivity of 1621 µA mM^−1^ cm^−2^ in the linear range of 0.0005–5.0 mM, a LoD of 200 nM and a fast response time of 3 s [43]. The sensor was applied for the determination of glucose in human serum working in 0.1 M KOH, i.e., the electrocatalytic process for glucose oxidation was mediated by the Cu(II)/Cu(III) redox couple, and the selectivity was investigated by recording the response current after adding some common interferents. The results showed that the sensor was highly selective, and this property together with the other excellent performances were related to the high conductivity of nanoporous Cu and the exceptional catalytic activity of CuO in alkaline solutions. Furthermore, since the micro device is integrated in a plane, it has great potential to be applied in portable sensors.

In the same year, another paper was published describing a device for glucose detection with analogous performances, fabricated as a sandwich-like nanocomposite, made of a vertical CuO nanowire array, grown homogeneously on both sides of a nanoporous thin film of copper which was produced by chemical dealloying process of a Cu-Zr-Al metallic glass. The sensor was applied to measure the glucose in blood serum samples, and the results obtained were in good agreement with the values of the biochemical analyzer used in hospitals with this aim [44]. The electrocatalytical activity of CuO can be further increased upon interaction with other materials facilitating the formation of oxide ion vacancies [45]. CeO_2_@CuO core shell nanostructures were synthesized and used to modify a graphite screen-printed electrode to construct a glucose sensor which displayed even higher sensitivity (3320 μA mM^−1^ cm^−2^) than the devices based only on CuO described earlier [46]. In particular, the authors demonstrated that the anodic peak current recorded for the CeO_2_@CuO-modified SPE was about seven times greater than the one observed for CuO. The excellent performance was attributed to the high electron transfer rate promoted by ceria (CeO_2_).

Another sensor was reported which was based on a CuxO-NiO nanocomposite fabricated by a novel procedure involving first the electrodeposition of porous Cu-Ni nanofilm on an anodized copper electrode, working at a cathodic potential at which hydrogen bubbles evolution occurs, and then the potentiodynamic oxidation of the metal nanofilm in a NaOH solution [47]. The analytical merit figures were a sensitivity of 1380 μA mM^−1^ cm^−2^, linear range from 0.04 to 5.76 mM, LoD of 7.3 μM, but the major drawback displayed by the electrode modified with CuxO-NiO nanocomposite was the loss of performance after exposure to high concentrations of glucose. On the other hand, it exhibited a stable signal for one months (only 6.3% decrease) and good selectivity and provided results for glucose levels in blood serum and urine samples in good agreement with the ones obtained using the methodology employed in the local hospital.

To further improve the performance of amperometric sensors exploiting the ability of various materials to act as redox mediators, carbon nanomaterials, such as CNTs and G, have been introduced as suitable electrode materials as they display a large surface area, high conductivity, good catalytic ability and provide effective charge transport channels [48].

A watchshake and flexible electrochemical sensor displaying great potential for personal and clinical use with portable electronics has been proposed. It exploits a 3D microstructure of highly conductive laser-induced graphene (LIG) on a polyimide substrate on which Cu_2_O nanocubes and Au NPs were electrochemically and chemically deposited, respectively. Taking advantages of the large surface area due to the wrinkles of LIG, its high conductivity and synergistic catalysis with the network of Cu_2_O and Au nanomaterials, both electron and mass transfer were exceptionally favored. Moreover, the device displayed good selectivity, and when applied to the determination of glucose in serum sample in the concentration range of clinical interest, i.e., 4–8 mM, it provided results in very good agreement with the glucose meter employed in the hospital (Figure 3b–e) [49].

**Figure 3 nanomaterials-13-01883-f003:**
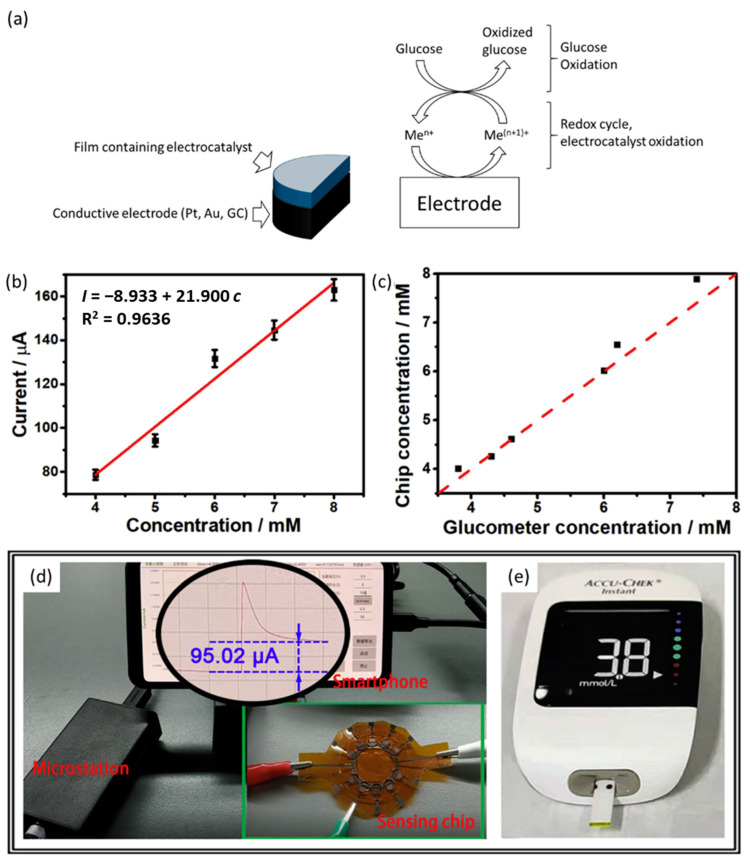
(**a**) Sketch of electrocatalytic mechanism for signal transduction in non-enzymatic glucose sensor (**b**) Calibration curve of the designed flexible chip in a blood serum sample. (**c**) Comparison of the test results between the proposed chip and the commercial blood glucose meter for glucose sensing in a blood serum sample. (**d**) Photograph of the glucose sensing system integrating with sensing chip, microstation, and smartphone to detect 4 mM glucose. (**e**) Photograph of the commercial blood glucose meter used to detect 4 mM glucose. Reprinted with permission from [49]. Copyright 2022 Elsevier.

Other metal oxides have been proposed as performant materials acting as redox mediators for the direct oxidation of glucose. A platform based on a hybrid nanomaterial made of Co(OH)_2_ NPs electrodeposited potentiostatically on a GCE modified with three-dimensional graphene frameworks was reported to construct an amperometric sensor displaying a better catalytic activity of the redox couple Co(III)/Co(II) in basic solution [50]. This characteristic was related to the high surface area and high conductivity for electron transfer of the 3D graphene framework and to its interconnected pores, which facilitate glucose and gluconolactone transfer from the solution to the electrode and vice versa.

The device displayed good selectivity toward AA, UA, DA, paracetamol and chloride ions which are present in the serum at concentrations of 0.1 M, and this property allowed it to be successfully applied for the determination of glucose in serum and urine samples. In addition, the glucose sensor could be re-used eight times if poured in NaOH solution, which made it suitable for point-of-care testing. The analytical parameters were: a sensitivity of 2410 μA cm^−2^mM^−1^, linear range from 2 μM to 1.4 mM, LoD of 0.67 μM and quick response (less than 3 s).

In 2022, Manjushree et al. published a paper dealing with a GCE modified with a composite made up of nanosheets of *β*-Ni(OH)_2_ wrapped with Ce_2_O_3_, which had been produced directly on the surface of reduced graphene oxide (rGO). The process for the electrocatalytic glucose oxidation via the nanocomposite under alkaline conditions was promoted by the redox couple Ni(III)/Ni(II), but the sensing activity was strongly enhanced by the synergistic impacts of rGO sheets and Ce_2_O_3_, which increased the rate of electron transfer. As a result the glucose assay sensitivity in serum samples was 33.1 μA mM^−1^ cm^−2^ [51].

#### 2.2.2. Metals, Alloys and Metal Compounds

Noble metals have been widely used as redox mediators thanks to their high electrical conductivity, excellent electrocatalytic performance and biocompatibility. On the other hand, they possess some drawbacks, essentially related to the high cost and low availability, and, above all, to the ease to be poisoned [52]. Nanoscopic electrode materials possess a very high active surface area that is significantly greater than the geometric surface area; this property is optimal for a kinetically controlled, surface bound reaction such as glucose oxidation [52].

A screen-printed carbon electrode was modified with micro-nano dual-porous gold, synthesized by electrodeposition using a hydrogen bubble as a template, in order to fabricate a compact and very sensitive device working at pH = 7.4. The sensor showed much higher currents at less positive potentials (−0.20 V vs. AgCl/Ag) for glucose oxidation when compared to an analogous SPE modified with gold nanoparticles. These features were ascribable to a higher efficiency both in electron and mass transfer, the latter being particularly favored by the interconnected electrolyte-filled dual porous network [53].

The sensor displayed a wide dynamic range from 25 µM to 16 mM of glucose concentration, a LoD of 25 µM (S/N = 3), and a sensitivity of 48.4 µA mM^−1^ cm^−2^ and, when applied to serum samples, provided data consistent with those obtained by an analyzer in clinical use.

An inkjet-printed paper electrode (PPE) with an optimized AuNPs aqueous nano-ink was proposed to develop a flexible and disposable electrochemical sensor for the detection of glucose in blood serum (Figure 4). It was used as a working electrode by applying cyclic voltammetry in PBS (pH = 7.0) and displayed a wide linear range from 0.05 to 35 mM, and a LoD of 10 µM. Additionally, the mechanical stability of the AuNP-PPE was remarkable, as confirmed through bending tests. Due to the properties of the inkjet printing methodology, the fabricated AuNP-PPE can be easily produced on a large scale for routine monitoring of glucose in biological samples [53].

Porous Pt has been reported to show high sensitivity and an excellent ability to be selective for glucose, due to the increased roughness factor of the electrode surface which could become 3000 times higher than the one displayed by smooth Pt electrodes [55]. To this aim McCormick and McCrudden employed the technique of electrodepositing an alloy of Pt/Cu on a disposable carbon SPE, and then of dealloying the less noble metal, to induce the nanoporosity, by applying 350 voltammetric cycles. The glucose determination was performed by amperometry in phosphate buffer (pH 7.4) at 0.4 V vs. Ag/AgCl. The sensor exhibited excellent stability, a linear range up to 13 mM and a rapid response time (<5 s) and was used to quantify the levels of glucose in blood serum samples with results in perfect agreement with the ones obtained using a commercial glucometer [56].

Compared with noble metals, transition metals such as Cu, Ni, Co and Fe show high electrocatalytic activity without displaying the above-mentioned drawbacks for noble metals, and, above all, they can be proposed to modify electrodes with the aim of developing devices at a lower price. Furthermore, one of the favorite methods for decreasing the overvoltage for glucose oxidation is the use of different metals and their compounds to prepare modified electrodes.

A highly sensitive electrochemical sensor for the simultaneous determination of dopamine, uric acid and glucose, using differential pulse voltammetry (DPV) in 0.1 M PBS (pH = 7.0), has been developed modifying a GCE with a nanocomposite made up of Cu and Ce nanoparticles, and carbon nanomaterials in the presence of Tween 20. The sensor exploited the high electrocatalytic ability for the oxidation of the three analytes of nano Cu, which was strongly amplified by the synergistic effect of the presence of the other nanomaterials. Linearity for glucose was obtained in the range 1–1000 µM with a LoD (S/N = 3) of 0.095 µM, and a good selectivity was demonstrated towards many inorganic and organic interferents, which allowed the accurate determination of DA, UA and glucose in blood serum samples [57].

A CuNi nanoalloy was easily electrodeposited on a GCE to design a sensor with higher electrocatalytic activity for glucose oxidation than the pure Cu or Ni nanoparticle-modified electrode. The characterizations showed that the CuNi nanoalloy possessed a rough specific surface area with a lot of electrocatalytic active sites of Cu and Ni, both acting as redox mediators in 0.1 M NaOH solution for glucose oxidation [52].

Due to the synergistic effect of the Cu^3+^/Cu^2+^ and Ni^3+^/Ni^2+^ redox couples the glucose sensor exhibited a low detection limit (0.02 μM), a fast current response (within 5 s), a linearity in the range of 0.05 to 35 μM along with a satisfactory analytical performance in human blood serum samples diluted with 0.1 M NaOH solution before performing the test.

As far as transition metal compounds are concerned, in particular those based on Ni, it can be stated that the literature was focused mainly on Ni oxides/hydroxides nanomaterials, but their electrocatalytic activity has been hampered by the semiconductor nature which limits their intrinsic electrical conductivity. To overcome this problem, Ni oxides/hydroxides have been often converted into nickel nitrides as they display better conductivity [58].

Furthermore, introducing carbon matrices into metal nitrides has proved to be an effective strategy to further boost their charge conduction and electrocatalytic performances. Liu et al. first proposed a sensor consisting of Ni nitride nanosheets dispersed on hollow and porous carbon fiber (PCF) through nitridation treatment of precursors. The characterization demonstrated that Ni_3_N-PCF had a 3D hollow and tubular architecture and a lot of meso/macropores contributing to build more mass transport channels. Moreover, the metallic nature of Ni_3_N and the excellent electroconductivity of PCF assured an improved electrode transfer kinetics, which led the fabrication of a modified GCE which can be considered as one of the best non-enzymatic glucose sensors in real human blood. The glucose concentrations were measured at +0.55 V vs. Ag/AgCl in 0.1 M NaOH solution in two wide linear ranges (from 0.001 to 1.75 mM and from 1.75 to 9.18 mM) with a very short response time (1.7 s) [57]. This work has opened a new approach for boosting electroanalytical abilities of Ni-based materials, even if transition metal nitrides suffer from poor electrochemical stability due to their easy oxidation, which can be overcome using appropriate carbon supports as just described. Deepalakshmi et al. synthesized a nickel cobalt nitride encapsulated in a nitrogen-doped graphene (NG) so as to produce a core–shell NiCo_2_N/NG nanomaterial which displayed electrochemical stability, thanks to the remarkable synergy between the core and shell materials, and highly sensitive and selective properties for glucose oxidation ascribable to the redox centers of Ni^3+^ and Co^3+^. It was used to modify a GCE working in a 0.1 M NaOH solution, and the resulting sensor exhibited a wide linear range (from 2.0 μM to 7.15 mM), a good sensitivity of 1803 μA mM^−1^ cm^−2^, a LoD as low as 50 nM at S/N = 3, and a rapid response (<3 s) [59]. The authors demonstrated also that the hierarchical core–shell superstructure of the nanomaterial minimized the aggregation of active NiCo_2_N, which enhanced the electroactive sites and improved the lifetime of sensors. In fact, the device was still operating after 45 days of contact with the atmosphere, displaying a reduction in the anodic current of 7.7% for 0.5 mM of glucose addition. The sensor was successfully applied to determine glucose in real human blood serum samples.

With the aim of making more efficient Cu-based electrocatalytical nanomaterials, a conductive copper complex, 7,7,8,8-tetracyanoquinodimethane, was directly grown on a Cu foam as a nanorod array and proposed as an efficient catalyst for amperometric glucose detection in alkaline conditions at 0.5 V vs. SCE [60]. The synthetic procedure assured a strong mechanical adhesion and a faster electron transfer due to a very small electron conduction distance between the nanorods and the current collector, which allowed the development of a sensor with a response time of 3 s, a wide linearity range (from 0.001 to 10.0 mM), a LoD value as low as 10 nM (S/N = 3) and a very high sensitivity of 29,687 μA mM^−1^ cm^−2^. UA, AA, fructose, urea, L-cysteine, lactose, glycine and glutamic acid were tested as interferents and the selectivity was optimal. The sensor was applied for the analysis of diluted serum samples. Due to its characteristics, the sensor can be considered an attractive low-cost device for clinical applications.

Still going in the direction of searching for other low-cost materials with both higher electrical conductivity and catalytic performance for glucose sensing than most metal oxides, a hierarchical three-dimensional Co phosphide decorated nanoporous copper, electrodeposited on a flexible copper clad laminate (Co-P/NP Cu/FCCL) was synthesized in order to fabricate a flexible electrode which displays great prospects in the development of wearable systems. The Co-P/NP Cu composite material exhibits the synergism between a phosphide with high catalytic activity and a nanoporous Cu structure which contributes to a high conductivity and a large surface area [42]. Working with 0.1 M KOH at a potential of 0.60 V vs. Ag/AgCl (saturated KCl), the device showed a sensitivity of 1818 μA cm^−2^ mM^−1^, a LoD of 378 nM, a fast response time of 4 s, a wide linear dynamic range (from 0.5 μM to 10.0 mM), a good anti-interference performance and a good reliability when applied to the analysis of glucose in serum samples.

Shakir et al. self-assembled Ag nanowires on ITO by exploiting a DNA template to fabricate a glucose sensor that operates in alkaline solutions with a sensitivity of 0.860 μA μM^−1^ cm^−2^. The authors validated the approach using recovery tests performed on human blood serum [61].

#### 2.2.3. Metal Organic Frameworks

Hierarchically porous nanomaterials are endowed with a large surface area and a lot of active sites which confer remarkably increased catalytic properties. At present, MOFs are a class of crystalline materials consisting of metal ions coordinated to organic ligands to form one, two, or three-dimensional structures which can be deposited on the substrate surface by means of many approaches, with the in situ growth involving a relatively simple and direct synthetic procedure which does not use any binder [62]. The electrochemical characteristics of MOFs depend on the type of metal and ligand and can be exploited to develop novel electrochemical sensors. The main drawbacks of MOFs are related to the low conductivity and stability issues, which can be overcome by adding carbon nanomaterials or employing highly conductive substrates [63].

In 2020, Shahrokhian S. et al. described, for the first time, a rapid and simple three-step in situ approach for the direct growth on a GCE of a Co-based MOF, i.e., Co_3_(BTC)_2_, starting from the electrodeposition of Co(OH)_2_ nano-flakes on rGO/GC followed by the fast conversion of the material to crystalline rectangular bar-shape structures of the MOF, simply pouring the modified electrode in a solution of the ligand H_3_BTC (1,3,5-benzene tricarboxylic acid [64]. The as-prepared Co_3_(BTC)_2_ MOFs were used to fabricate a sensing platform for glucose in an alkaline solution which confirmed that the CoOOH/CoO_2_ couple is the redox mediator for electro-oxidation. The sensor displayed two wide linear concentration ranges from 1 μM to 0.33 mM and from 0.33 mM to 1.38 mM with sensitivities of 1792 μA mM^−1^ cm^−2^ and 1002 μA mM^−1^ cm^−2^, respectively, and a LoD of 0.33 μM (S/N = 3). Furthermore, the MOF-based electrode exhibited high selectivity against interferents such as UA, AA and DA and good poisoning resistance against chloride ions. The method of the standard addition was used for determining glucose in serum samples, but also in human urine and saliva samples, which resulted in agreement with the value obtained from a commercial glucometer.

To improve the efficiency of glucose sensors based on core–shell bimetallic nanoarrays, the synthesis of self-supporting and well-ordered nanostructures plays a crucial role. These ordered nanoarrays electrodes have been synthesized using a direct growth of a MOF (Co-ZIF-67 compound) on the outer surface of Cu(OH)_2_ nanorods produced after anodization of a Cu foam. Then, the core–shell CuOx@Co_3_O_4_ nanowires could be easily obtained through the decomposition and oxidation of Cu(OH)_2_ nanowires and porous ZIF-67 layer, occurring during a pyrolysis step [65]. The as-obtained nanowires displayed an almost unchanged morphology, i.e., they were highly aligned on the Cu foam, perfectly connected to each other, and their surface had become much more porous. This particular structure of the bimetal oxide catalyst possessed a very high number of catalytic active sites which are also more accessible to the substrate and led to a significantly increased catalytic activity. In fact, the glucose sensor exhibited a great sensitivity (27,778 μA mM^−1^ cm^−2^ in the range of 0.1 to 1300.0 μM), a LoD equal to 36 nM (S/N = 3) and fast response time (~1 s); furthermore, it also displayed satisfactory selectivity, reproducibility and long-term storage stability. It was successfully applied to measure the levels of glucose in human blood serum, recording the amperometric response after injecting 50 μL of sample.

As already stated, combining metal alloys with carbon materials is an effective way of synthesizing electrocatalysts with superior activity by increasing the number of available active sites, and providing effective charge transport channels due to the porous structure and large surface area. To this aim, Wang et al. synthesized, via hydrothermal method, a MOF using a solution of *p*-phthalic acid and Co and Ni ions. After a pyrolysis process under a nitrogen atmosphere, a core–shell NiCo/C composite was obtained. NiCo NPs were coated with graphitized carbon, which avoided the aggregation of NiCo alloy NPs and improved the conductivity of the material [62]. It was employed to modify a SPE, which was applied for electrocatalytic glucose oxidation. In the linear range 0.5 μM–4.38 mM, the sensor showed a sensitivity of 265.53 μA mM^−1^ cm^−2^ with a LoD = 0.2 μM (S/N = 3). It also had a good repeatability, relatively long-term stability (>30 days) and selectivity and was able to accurately determine the glucose concentration in human serum.

Another paper described the conversion, through an etching process, of the hydrophobic ZIF-8 crystal, a zeolite-type MOF, into a hydrophilic hierarchically porous nanoflowers structure, whose pores were transformed from a micro to meso and macro sizes [66]. This peculiar feature was exploited to immobilize Cu NPs, acting as redox mediators, for glucose oxidation in alkaline solution. The composite, suspended in water, was drop cast on a GCE and the modified electrode exhibited a more intense redox wave for glucose oxidation when compared with the electrode modified with Cu NPs loaded on the untreated ZIF-8. When standard solutions of glucose were tested, the sensor exhibited exceptional sensing performances with a linear range from 5 μM to 3 mM, a LoD of 1.97 μM (S/N = 3), a sensitivity of 1594.2 μA mM^−1^ cm^−2^, and high selectivity. It was employed to determine glucose in serum samples and the current density of the oxidation peak displayed a linear relationship with the concentration in the range of 0 to 10 mM, so it was very promising for real samples analysis.

The main analytical features of all the glucose (bio)sensors described in Section 2 are shown in Table 1.

In summary, the enzymatic devices more recently described in the literature, especially those based on noble metals NPs or nanoporous materials, exhibit an increased stability, due to the better adsorption ability of such nanomaterials, making their re-usability possible. In such a way, the high cost of the sensor related to the enzyme presence is largely addressed. The same holds also when the biosensors exploit carbon nanomaterials, especially graphene, but in such a case, in addition to the high specific surface area favoring the enzyme immobilization, the presence of G makes the DET occurrence easier so as to pave the way for the development of third-generation electrochemical biosensors. In such a way, the production of self-powered biosensors by employing two enzymatic electrodes should rapidly lead to a real marketing of portable devices.

The non-enzymatic electrochemical glucose sensors, reported in the last five years within the literature, have especially proposed the use of hybrid nanomaterials and nanoporous supports. Employing different metals and/or their compounds to prepare modified electrodes aims to decrease the overvoltage for glucose oxidation in order to overcome the issues related to selectivity. The nanoporosity of the supports or of the modifiers allows for a greatly enlarged electrochemically active surface area, which is generally advantageous in terms of sensitivity. When the electrocatalysts are metal oxides or MOFs, the main drawback related to their low conductivity is generally overcome by adding carbon nanomaterials.

In conclusion, almost all the devices just described offer more promising tools for diagnosing diabetes and monitoring daily blood glucose levels.

## 3. Other Body Fluids

The accurate detection of glucose without hysteresis is one of the most important clinically unmet needs for glucose monitoring of the body fluids to track blood glucose levels.

Blood sugar monitoring devices are commonly used as medical devices in the management of diabetes. A typical example of a self-monitoring device is the Accu-Check Guide, released by Roche in August 2016. Other systems available to the market are those allowing a continuous glucose monitoring of the interstitial fluid, offering a more thorough understanding of glucose levels and trends than those of traditional devices. Three examples of devices, currently approved by the U.S. Food and Drug Administration, are the Dexcom G4 Platinum Professional, Medtronic iPro2 Professional CGM and FreeStyle Libre Pro System from Abbott. The last one allows the monitoring of the interstitial glucose level every 15 min for up to 14 days, using a small sensor attached on the arm.

In addition to the quite invasive analysis of blood and interstitial fluid, other easily accessible biological fluids such as ocular fluid, sweat, breath, saliva or urine have been investigated as alternative samples for non-invasive continuous monitoring. Non-invasive CGM can be achieved by fabricating sensors integrated in contact lenses, watches, tattoos, and patches.

Typical glucose concentrations for healthy and diabetic people in physiological fluids are reported in Table 2. It is worth noting that the expected glucose concentration in sweat, saliva and tears is very low if compared to that in blood, IF and urine, making the development of a sensor more challenging, since it must be very sensitive and able to work with small sample volumes. It is worth noting that a wound secretes up to 0.9 μL cm^−2^ min^−1^ [67] and the sweating rate ranges from 0.12 to 0.99 μL cm^−2^ min^−1^ [68], while the tears volume is about 7 μL [69,70].

In the following paragraphs, the most relevant devices using nanomaterials in their structure for the analysis of biofluids other than blood are summarized and grouped according to the biofluid in which the authors have proposed to employ the device.

### 3.1. Urine Analysis

The most recent devices proposed for the analysis of glucose in urine are non-enzymatic systems, working in basic environments, which exploit the electrocatalytic features of transition metals oxides, hydroxides or complexes.

Dai et al. fabricated a device based on Co_3_O_4_/PPy (polypyrrole) core–shell nano-heterostructures electrodeposited in two steps on a porous nickel-foam, which exhibited a fast response time (12 s), a low LoD (0.74 mM), and high selectivity. The sensor’s high sensitivity for glucose detection was ascribed to the synergistic effects among Co_3_O_4_, PPy and Ni foam, and its selectivity to the electrostatic repulsion between the negative charges of Co_3_O_4_ and the tested interferents (D-fructose, AA, UA, and hydrogen peroxide) [71].

It is worth noting that a Cu_x_O-NiO nanocomposite and Co(OH)_2_ NPs on three-dimensional graphene frameworks have been already described in Section 2, as they were employed as sensors for the detection of glucose blood in addition to urine [47].

The capability of Cu oxides systems to electrocatalyze glucose oxidation was also exploited by Viswanathan et al. who synthesized Cu/CuO/Cu(OH)_2_ in the presence of DA. Dopamine has a key role in the electrode preparation since, on one hand, it reduces Cu(II) to Cu(0), thus improving the conductivity of metal oxides/hydroxides and, on the other hand, it is in situ converted to its polymeric form, that wraps around the electrocatalyst, therefore increasing the biocompatibility of the resulting material. The composite modified electrode exhibited a sensitivity (223.17 μA mM^−1^ cm^−2^), a LoD of 20 μM, a linear range up to 20 mM and a short-response time (<3 s), along with excellent selectivity, reproducibility and stability [72].

A carbon paste electrode modified with MWCNTs and N-salicylaldehyde, N′-2-hydroxyacetophenon-1,2-phenylene diimino Nickel(II) complex was proposed by Rezaeinasab et al. for the determination of glucose in blood serum and urine solutions. Under the optimum conditions, the sensor displayed a wide range of response with the occurrence of two linear ranges, i.e., from 5 to 190 mM and from 210 to 700 mM, and a LoD of 1.3 mM. The device exhibited some advantages such as low cost, high stability and selectivity, high reproducibility and ease of preparation [73]. In another report, SPEs were decorated with carbon nanohorns, and the surface was further modified with nickel-cobalt sulfide nanosheets. The resulting sensors exhibited two distinct linear ranges of 0.001–0.330 mM and 0.330–4.53 mM with sensitivities of 1842 µA mM^−1^ cm^−2^ and 854 µA mM^−1^ cm^−2^, respectively, and they could be employed for glucose determination in human blood serum, urine and saliva samples without any sample pre-treatment [74].

Hernández-Ramírez et al. have proposed a sensor based on Fe_2_O_3_ nanoparticles as electrocatalytic modifiers immobilized on carbon paste electrodes operated by DPV. Three linear ranges were identified at alkaline pH: 0.015–1 μM, 1–100 μM and 30–700 μM (sensitivity of 0.041 μA μM^−1^), and the sensor selectivity was demonstrated for AA, UA, lactose, caffeine and paracetamol. Glucose detection was carried out in commercial beverages and human urine samples [75].

A gold electrode modified with a composite prepared by mixing zinc oxide rods and ruthenium-doped graphitic carbon nitride was recently reported which showed a sensitivity of 346 μA mM^−1^cm^−2^ over a broad linear range of concentration (2–28 mM). Furthermore, the sensor displayed fast response (3 s), a very low LoD (3.5 nM) and was reusable. Its suitability to real-life applications was proved through the analysis of human blood, serum and urine samples [76].

Similarly, a Ag-doped ZrO_2_ nanocomposite combined with graphene-based mesoporous silica was evaluated for the electro-oxidation of glucose using cyclic voltammetry in PBS at pH 7.4 and commercial urine [77].

Au containing nanocomposites were also reported for the non-enzymatic detection of glucose in urine samples. In one case, disposable laser-induced graphene electrodes were modified with core–shell Au and CuO NPs (Au@CuO) exhibiting electrocatalytic activity in alkaline medium. Au@CuO/LIG showed great stability and exhibited a detection range from 0.005 to 5.0 mM [78]. In another case, a simple synthetic procedure to obtain gold@nickel oxide nanodentrites microarrays was proposed for the electrochemical sensing of glucose and lactate in human serum and urine samples. The sensor, operating at 0.5 V, exhibited a wide linear range from 10.0 μM to 5.0 mM of glucose, a LoD of 100.0 nM, and a high selectivity for glucose which was verified by performing the detection in the presence of 10-fold high concentration of interferences such as DA, UA, AA, mannose, maltose, galactose, paracetamol and Ca and Mg ions [79].

A non-enzymatic electrochemical sensor was recently reported exploiting GO chemically reduced with a a phyto-extract (Aconitum heterophyllum plant root) which was drop cast on Fluorine-Doped Tin Oxide. Interestingly, the electrocatalytic properties were ascribed to the immobilized phyto-extract and selective glucose sensing, with a sensitivity of 61.76 µA mM^−1^ cm^−2^ at 0 V vs. Ag/AgCl, was carried out in citrate phosphate-buffered solution. Thanks to the linear response (0.3–33.3 mM) that lies within a diagnostically relevant range, the authors suggest a possible application for the analysis of blood or urine analysis [80].

Finally, Li et al. described a smart diaper for in situ detection of major urinary metabolites (glucose, reactive oxygen species, UA) and electrolytes (K^+^ and Na^+^ ions). It was based on integrated multiplex electrochemical sensors which were fabricated using CNTs, functionalised with ion-selective membranes, enzymes, or Pt NPs in order to achieve the desired selectivity [81]. In the case of glucose, the sensor was fabricated by dropping a mixture of GOx/BSA/GA on a Pt/CNT/Au working electrode, employing a Nafion membrane to increase stability.

The main analytical features of all the glucose sensors described for urine analysis are shown in Table 3.

### 3.2. Sweat Analysis

Patch-type wearable glucose sensors can be attached to the human body to continuously and non-invasively analyze the glucose levels in sweat. The detection of glucose in sweat is particularly interesting due to the abundance and easy accessibility of this biofluid in the human body, but it is particularly challenging due to the low levels of glucose.

A device based on wrinkled, stretchable nanohybrid fibers (WSNF) made of reduced graphene oxide–polyurethane (rGO/PU) partially covered with Au nanowrinkles, to increase the surface area, was described by Tan Toi et al. Thanks to the synergistic effects between the Au nanowrinkles and the rGO supporting matrix, the sensor electrocatalytic activity was increased with a consequent sensitivity (140 μA mM^−1^ cm^−2^). Furthermore, the device exhibited a LoD of 500 nM, high selectivity against interferents (tested AA, UA, lactate and NaCl), stability at ambient conditions, ability to work in neutral solutions, and to perform CGM (Figure 5). The authors demonstrated the operativity of the device by attaching it to the forehead of two healthy subjects and continuously measuring glucose levels in sweat by means of a portable electrochemical analyzer wirelessly connected to a smartphone. A good correlation between the glucose concentration in sweat and the one in blood, either before or after food ingestion was established [82].

Recently, an ultra-compact glucose tag with a footprint and weight of 1.2 cm^2^ and 0.13 g, respectively, has been developed for sweat analysis. Thanks to near field communication-based wireless power transmission and a compact antenna, the device can work in a battery-less fashion. For glucose sensing, a printed carbon electrode was electrochemically modified with PB and coated with a chitosan/GOx/Au NPs mixture, exhibiting a LoD of 24 μM, a limit of quantification (LoQ) of 74 μM and sensitivity equal to 1.27 μA cm^−2^ mM^−1^. The device capability when attached to the body, the sweat collection, and glucose measurement were demonstrated through in vitro and in vivo experiments [83].

In another recent report, tannic acid-3-aminopropyltriethoxysilane (TA-APTES) coatings were prepared on an activated carbon cloth (aCC) and, after in situ reduction of HAuCl_4_, AuNPs/TA-APTES/aCC electrodes were obtained. Once GOx was immobilized onto the electrode surface, the biosensors displayed a sensitivity of 72 μA mM^−1^ cm^−2^ in the range 0.01–18 mM glucose and were tested in sweat samples showing good stability [84].

A flexible enzymatic sensor was recently reported based on a laser-scribed graphene with a 3D porous structure and good conductivity, which was modified with a hydroxyethyl cellulose film and Pt NPs to take advantage of the hierarchical and porous graphene architectures combined with the electrocatalytic activity of Pt NPs. After GOx immobilization, the biosensor exhibited a sensitivity of 69.64 μA mM^−1^ cm^−2^ and a limit of detection of 0.23 μM, with a detection range of 5–3000 μM, thus covering the glucose range in sweat. Furthermore, the authors demonstrated that the same sensor (without GOx), when functionalized with polyaniline (PANI), displayed high pH sensitivity (72.4 mV/pH) in the linear range of pH 4–8. The dual-functional flexible sensor was applied to the real analysis of human perspiration during physical exercise [85].

In another example, a flexible biosensor for sweat glucose detection was designed comprising a gold electrode, PB, a graphene sponge (GS), CS and GOx, where the large specific surface area and the cross-linked porous structure of GS promoted the adsorption of a large amount of the enzyme. The biosensor exhibited a sensitivity of 1790 nA mM^−1^ cm^−2^ and a detection limit of 2.45 μM (S/N = 3) at a low applied potential (0.075 V, vs. Ag/AgCl) in a linear range from 8.17 to 1000 μM. The device was tested for glucose detection in real sweat, which was collected from volunteers playing table tennis, obtaining a good correlation with a commercial kit [86].

Garg et al. proposed a hydrogel-based device able to monitor human-body motion and glucose concentration in sweat. The investigation focused on the development of a hybrid hydrogel nanocomposite, constituted of polyvinyl alcohol (PVA) as the primary matrix, PANI as a conductive secondary polymer and thermally exfoliated GO as the conductive reinforcement as well as the template for the deposition of PANI chains. The hydrogel possessed high conductivity (0.14 S m^−1^), superior mechanical strength (up to 7.7 MPa) and toughness (up to 7.48 MJ m^−3^) and the preparation method was fast, low cost and easily scalable. A proof of concept for a glucose amperometric sensor was designed immobilizing GOx at the hydrogel surface. Thanks to the hydrogel nano-porous structure, the sensor operated in a concentration range 0.2 μM–10 mM with a LoD equal to 0.2 μM [87].

In analogy to this work, rGO and Au NPs-modified screen-printed gold electrodes (rGO-Au/SPGE) with immobilized GOx were studied for enzymatic glucose sensing using a gel, instead of a traditional liquid electrolyte, which should adhere better to the skin and promote the accumulation of body fluids on dry skin surfaces. The as obtained biosensor maintained a stable sensing performance even after 1000 bending cycles and exhibited two detection ranges, i.e., 1.25–850 µM and 0.85–7.72 mM, with sensitivities of 53.7 and 27.4 μA mM^−1^ cm^−2^, respectively [88].

Moreover, another non-invasive sweat glucose sensor was recently reported, based on hydrogel patches for the rapid sampling of natural perspiration from the hand. The biosensor comprised a PB-doped poly(3,4-ethylenedioxythiophene) nanocomposite (PB-PEDOT NC), a GOx/CS layer and a Nafion film, and was employed for the long-term measurements of glucose in sweat from human subjects consuming food and drinks [89].

A. Müsse et al. developed a nanolithography and nanoimprinting process for the fabrication of a simple microfluidic system with inlet, outlet and a circular chamber, thus facing the problem of analysing low sample volumes, such as in the case of sweat analysis. The device provided a uniform fluid flow over a three-electrode system employing Au as working and counter electrodes and Ag as pseudo-reference electrode, which were evaporated on a flexible polystyrene foil. The sensing layer was prepared by coating the Au electrode with a cysteine self-assembled monolayer and GA, BSA and glycerol were employed for GOx immobilization. The sensor, operating in PBS solution at 0.7 V, showed a linear range from 0.025 mM to 2 mM, whereas the operational range was wider (from 0.025 mM to 25 mM) maybe due to the addition of BSA and glycerol as stabilizing agents. The sensitivity was 1.76 μA mM^−1^ cm^−2^ and the LoD was 0.055 mM, making the sensor suitable for the detection of glucose in sweat and body fluids other than blood. Due to the anodic operative potential, the device is expected to suffer from the interference of all the oxidizable species present in the fluid under investigation, and thus a Nafion layer was added in view of real application [90].

Myndrul and co-workers proposed electrodes modified with ZnO tetrapods (TPs) and transition metal carbides (MXene) nanoflakes for on-body qualitative enzymatic glucose monitoring in sweat. The device took advantage of the very high active surface area provided by the large pods and highly porous 3D interconnected networks of ZnO TPs, and of the outstanding metal-like conductivity of MXene. The ZnO TPs/MXene was synthesised in a three-step manufacturing process, including the synthesis of MXene and ZnO with the following decoration of ZnO TPs with MXene nanoflakes. The sensor was stable for a period of 10 days and exhibited a sensitivity of 27.87 μA mM^−1^cm^−2^ and 29.88 μA mM^−1^ cm^−2^ in PBS and artificial sweat, respectively, and a LoD of 21 μM. Moreover, since the device worked at a potential of −0.24 V vs. Ag/AgCl, it was highly selective since none of the usual interfering agents affected its response. The real-time monitoring of post-meal glucose levels in sweat had the same trend as measured with a conventional blood glucometer under similar conditions [91].

All the previously described devices exploited GOx for glucose detection and worked in neutral environment. A non-enzymatic sensor based on the composite CeAlO_3_-carbon nitride was proposed by Rajaji and co-workers. The authors presented a novel green synthesis of CeAlO_3_ polycrystalline powder by a hydrothermal method from CeO_2_ and Al_2_O_3_ in deep eutectic solvents and prepared CN starting from melamine, by fractional thermal polymerization method. The final composite, which was deposited on a GCE, was then obtained by mixing CeAlO_3_ and CN powders. The sensor showed high electrocatalytic properties for glucose sensing in 0.1 M NaOH solution due to the synergistic effects of CeAlO_3_ and CN. The LoD was 0.86 nM, and the linear concentration range from 0.01 to 1034.5 μM. Actually, two linearity ranges were observed, the former from 0.01 to 180 μM with a sensitivity of 68,718 μA mM^−1^cm^−2^ and the latter from 200 to 1034.5 μM with a sensitivity of 81,408 μA mM^−1^cm^−2^. The modified electrode exhibited negligible interference from UA, AA, DA, nitrate, nitrite, hydrogen peroxide, folic acid, paracetamol, epinephrine, ciprofloxacin and norfloxacin. It was used to successfully measure glucose concentration in diluted samples of sweat and saliva but, since it operated in basic environments, it was not suitable for a direct attachment to the skin for real time sweat detection [92].

In a recent paper, a CNT/CuO NC was fabricated to improve the direct electron transport to the electrode surface during electrocatalytic glucose oxidation. With a LoD of 3.90 µM and a sensitivity of 15.300 × 10^6^ μA mM^−1^ cm^−2^ within a linear range of 5–100 µM, the developed sensor showed a high selectivity to glucose in the presence of various bio-compounds found in sweat [93].

Asen et al. developed non-enzymatic sensors based on SPE modified with GO nanosheets supporting flower-like Au nanostructures. When tested in artificial sweat, two linear ranges from 160 nM to 82 μM and from 160 μM to 5 mM were identified, which coincided with the glucose in healthy and diabetic individuals, and recovery tests in sweat samples were carried out [94].

Electrocatalytic properties can also be boosted through the design of bimetallic nanocomposites. An enzyme-free sensor was recently reported combining nickel-samarium nanoparticles-decorated MXene layered double hydroxide (Mxene/Ni/Sm-LDH), which possessed electrocatalytic activity for glucose oxidation. Two linear ranges were identified in DPV, i.e., from 0.001 to 0.1 mM and from 0.25 to 7.5 mM with a LoD of 0.24 μM (S/N = 3), and the sensor was proven promising for glucose detection in human sweat [95].

In another report exploiting MXene-based nanocomposites, a flexible non-enzymatic electrochemical sensor for continuous glucose detection in sweat was fabricated starting from the synthesis of Pt/MXene catalyst by the hybridization of Pt nanoparticles onto MXene (Ti_3_C_2_Tx) nanosheets; later, the Pt/MXene was immobilized by means of a conductive hydrogel to enhance the stability of the sensor. Moreover, a microfluidic patch was attached for sweat collection onto the flexible sensor, which could detect the sweat glucose change with the replenishment and consumption of energy by the body. A similar trend of glucose levels was observed in the blood [96].

Zha et al. reported on a non-invasive flexible glucose sensor where NiCo-MOFs were exploited as micro-supercapacitor and sensing material. The sensitivity of the prepared bimetallic product was 1422.2 μA mM^−1^ cm^−2^, and upon the integration of the glucose sensor with a micro-supercapacitor on a flexible PET substrate, the authors suggested the potential application for non-invasive sweat glucose detection [97].

A GCE coated with nitrogen-doped carbon seaweed-like nanosheets containing Co and Cu was reported, which exhibited a sensitivity of 1489 μA μM^−1^ cm^−2^, a LoD of 0.34 μM and a wide linear range (1–1.25 mM), and was successfully tested with artificial sweat [98]. The sensor demonstrated good anti-interference, reproducibility and ultra-long stability, so it has promising application prospects for glucose detection in sweat.

An ultrathin 2D nanosheets array composed of trimesic acid (H_3_BTC)-based bimetal (Ni and Co) MOFs and carbon cloth was fabricated with a strict control of morphology. The resulting NiCo-BTC/CC sensor showed a high sensitivity of 2701.29 μA mM^−1^ cm^−2^ in the linear range 5–205 μM, with low detection limit (0.09 μM, S/N = 3). Additionally, in this case, the results obtained during glucose detection in sweat with the proposed device were consistent with those of a commercial glucometer [99].

A similar strategy was followed to design a wearable electrochemical sweat sensor based on a Ni–Co MOF nanosheet coated Au/polydimethylsiloxane (PDMS) film. The flexible sensor exhibited a wide linear range from 20 μM to 790 μM and a high sensitivity of 205.1 μA mM^−1^ cm^−2^. Moreover, the device was endowed with a sweat-absorbent cloth to form a wearable sweat glucose sensor, which could be used for CGM in human sweat for one day [100].

Interestingly, an enzyme-free sweat glucose sensor was also obtained with Whatman filter paper. The authors fabricated pencil-drawn electrodes on the filter paper using a dual-step writing process with a graphite pencil, and then one electrode was decorated with a sensing layer of Cu NPs ink and the other was drawn with a Ag conductive ink pen to form the reference electrode. The large surface area of Cu NPs allowed a fast electro-oxidation of glucose in a strongly alkaline medium, and the ink deposition made it easy to permeate the glucose-sensitive copper nanoparticles deep into the porous paper fiber. The paper-based sensor exhibited a sensitivity of 2691.7 μA mM^−1^ cm^−2^, a LoD of 0.5 μM, a linear range of 1.2–40 μM, and a fast response time of ~1.5 s, thus displaying potential for glucose detection in human sweat [101].

Thanks to the signal amplification, low energy consumption and ease of miniaturization, transistor-based sensors have effective potential for the design of wearable devices. A flexible gel electrolyte graphene transistor was reported for glucose detection, consisting of a gate electrode decorated with Au NPs modified reduced graphene oxide (AuNPs/rGO) nanocomposites, and a channel made of a graphene monolayer thanks to its optimal electron mobility. A glycerin gel ensured good adhesion to the human skin and provided a stable testing environment where the biofluid could be easily collected. The transistor-based sensor exhibited low LoD (10 nM), wide concentration range (10 nM–25 mM) and low operating voltage (0.5 V), which made it suitable for real application in sweat analysis during actual wearing [102].

Natural materials can also be used to build nanostructured glucose sensors. A self-assembly of glucuronic acid (GlcA)-coated Au nanoflowers on flexible carbon cloth was recently proposed for non-enzymatic electrochemical glucose detection. The presence of GlcA leading to more exposed nanoedges than the normal AuNFs/CC and, the unique {110} facets of Au atop the amino group grafted CC guaranteed the outstanding performance of the sensor. With a linear range from 5 to 42 mM, a low LoD (5 μM, S/N = 3) and a high selectivity, the sensor was successfully applied for the detection of glucose in human sweat samples and beverages in real time [103].

The main analytical features of all the glucose (bio)sensors described for sweat analysis are shown in Table 4.

### 3.3. Tears and Saliva Analysis

Tears are maybe the most challenging biofluid to test: first of all, due to their small volumetric size and low glucose concentration, and secondly, because there is no full agreement in the literature on the existence of a correlation between glucose levels in blood and tears. The detection of glucose in tears could be realized thanks to the use of smart contact lenses integrating a miniaturized glucose sensor. In 2014, Verily Life Sciences, in partnership with Novartis’s eye care division Alcon, announced it was developing such smart contact lenses, but five years later they affirmed that the project had been put “on hold” [90]. Only very recently, few works concerning the development of proof-of-concept glucose sensors that are compatible with tears analysis and/or were tested in artificial tears have been reported.

For instance, a bimetallic nanocomposite Fe_x_Co_y_O_4_ was synthesised on rGO, where the synergistic effect of the two metals proved to be beneficial to obtain enhanced electrocatalytic properties. The authors chose a flexible SPCE to build a non-enzymatic glucose sensor, which exhibited a LoD of 0.07 μM, and a sensitivity of 1510 μM cm^−2^ mA^−1^. The device was successfully tested in untreated human tears mixed with 0.1 M NaOH [104]. Additionally, Cu oxide nanoparticles and nanostructures have been proposed as electrode modifiers for building enzyme-free glucose sensors with analytical performances that are compliant with tears analysis. MWCNTs were decorated with CuO NPs, which were synthesised using basil seed mucilage as reducing, capping, and stabilizing agent. The sensing performance of a SPCE modified by the nanocomposite toward the glucose oxidation was characterised by a sensitivity of 1050 μA mM^−1^ cm^−2^ in the range 5.0–620.0 μM and a LoD of 1.7 μM. The use of a negatively charged acidic polysaccharide coating, which is similar to Nafion, decreased the possibility of interference by anionic substances, such as lactate, AA and urea [105]. CuO/Cu(OH)_2_ nanostructures on graphene paper were prepared by thermal and nanosecond pulsed laser dewetting of a nanometric thickness CuO layer. The procedure led to the formation of nanostructures of different morphology, composition and dimensions. Amperometric detection of glucose was carried out at 0.4 V vs. SCE, showing a linear range from 50 μM to 10 mM and a LoD of 7 μM [106].

In another paper, a self-powered organic electrochemical transistor (OECT)-based multiplexed sensor was reported for future integration within contact lenses. For enzymatic glucose detection, GOx was deposited on the Pt gate electrode mixed with chitosan, while an ion-selective membrane was deposited on the channel of the Ca^2+^ OECT device. It was tested in human tears collected before and after breakfast and the sensing signals were wirelessly transmitted to a laptop through a near filed communication unit [107].

A very interesting work by Kim and co-workers described the development of smart contact lenses for glucose monitoring (Figure 6a), where the electrochemical sensor was based on bimetallic Au and Pt nanocatalysts modified with hyaluronic acid (HA-Au@Pt BiNCs) and where GOx was immobilized in a nanoporous hydrogel (Figure 6b). After the GOx catalyzed oxidation of glucose, hydrogen peroxide was rapidly decomposed on the surface of HA-Au@Pt BiNCs and the resulting current measured (Figure 6c). HA-Au@Pt BiNCs facilitate nanoparticle-mediated charge transfer in nanoporous hydrogels, resulting in sensitivity (180.18 μA mM^−1^ cm^−2^), fast response time (3.6 s), and low LoD (0.6 μM). In addition, the nanopores in the hydrogels enable the reversible and continuous monitoring of glucose concentration with a fast diffusion rate of reaction species and a rapid swelling rate. The authors demonstrated that the smart contact lenses were able to accurately monitor increasing and decreasing blood glucose levels with 94.9% acceptable data in diabetic (*n* = 3) and normal (*n* = 3) rabbits [108].

Finally, we want to discuss one of the few examples of a sensor based on nanomaterials and aptamers that works in saliva. Liu et al. proposed a wireless chip connected to a smartphone for dual sensing of insulin and glucose in saliva to achieve the diagnosis of diabetes/prediabetes. The transduction strategy is based on two thiolated aptamers that selectively bind glucose or insulin and are immobilized onto Au NPs decorated SPCE. The authors demonstrated the reliability of the proposed sensor by obtaining high recovery during the analysis of spiked saliva [19].

### 3.4. Ulcers and Wounds Analysis

One of the fatal outcomes of diabetes is diabetic foot ulcer which, once chronic, may lead to amputation.

Mathur et al. faced the problem whilst developing a multimodal enzymatic impedimetric sensor to monitor glucose and L-tyrosine, two biomarkers whose level becomes elevated in patients suffering from diabetes and diabetic foot ulcers. The proposed device, realized on paper, consisted of two working electrodes and a counter electrode made by carbon ink. One working electrode is modified with *α*-MnO_2_-Graphene Quantum Dots/tyrosinase hybrid, while the other is coated with *α*-MnO_2_-Graphene Quantum Dots/GOx hybrid, for L-tyrosine and glucose determination, respectively. Electrochemical impedance spectroscopy in the presence of the redox probe Fe(CN)_6_^3−/4−^ was employed for the quantification of glucose and L-tyrosine, within a concentration range of 3–50 mM and 1–500 μM, respectively, using a sample volume of approximately 200 μL. The impedance response exhibited a linear relationship over the whole analyte concentration range with a LoD of 3 mM and 0.3 μM for glucose and tyrosine, respectively, and a shelf life of about 1 month. The sensing strategy was also transferred to arduino-based device applications by interfacing the device with miniaturized electronics, thus demonstrating its high potential in the field of point-of-care applications [109]. In another recent example, an adhesive-free, permeable and multiplex sensor system was designed to obtain a biointegrated platform emulating the native epidermal mechanics and comprising a multiple biosensor array for the continuous monitoring of inflammatory biomarkers such as lactate, glucose, pH, oxygen and wound temperature, which correlate to the wound healing status. For the enzymatic sensors, GOx and lactate oxidase were immobilized with a CS and single-walled CNTs suspension on a PB mediator layer, which was electrochemically deposited by CV on Au electrodes patterned on PDMS fibers. The modified Au electrodes were biased at 0 V vs. Ag/AgCl, displaying a linear response range of 0–8 mM and a sensitivity of 2.98 μA mM^−1^ cm^−2^. The sensor selectivity was tested against AA, UA and KCl, giving negligible signals compared to glucose, thanks to the low applied voltage. The platform was tested in vitro, using a PBS solution, and ex vivo, using human wound exudates. The results suggested that the sensors should be calibrated in a matrix that simulates exudates, due the different viscosity, bio-fouling, pH, chloride concentration and solution conductivity which may vary the sensor response [110].

## 4. Implantable Devices

Implantable devices for continuous glucose monitoring still suffer from several problems such as interferences, mechanical friction and biocompatibility. The active substances present in the interstitial fluid can directly react with the electrodes, causing errors in the recorded signal. Mechanical friction on the electrode layer, which is induced when devices are worn, leads to electrode failure; in addition, the biocompatibility and cytotoxicity of invasive electrodes must be still investigated.

Cai and co-workers developed a biosensor for in vivo monitoring, successfully employed for glucose levels detection in rats in a 24 h implantation period. Before implantation, several experiments were performed to evaluate the biocompatibility of the device and its feasibility for in vivo application. The device consisted in a flexible double-sided screen-printed system, where the working electrode was made of a PB-doped carbon ink modified with nanostructured PANI/GOx. A PU membrane was employed as a protective coating to widen the linear range of response (0–12 mM) and increase the biocompatibility. The biosensor, thanks to the presence of PB, was able to operate at 0 V to reduce hydrogen peroxide produced by the enzymatic reaction, thus displaying a high selectivity. The sensitivity of the device was not so high (16.66 μA mM^−1^ cm^−2^) as the one typical of traditional metal needle electrodes, but the response was considered adequate for the detection of blood glucose [111]. The authors compared the data obtained by their implanted sensor to the blood glucose concentration, finding almost the same trend but with a delay of about 10 min, as the device measures the glucose level in the ISF [112].

Regiart et al. proposed a dual biosensor for the simultaneous in vivo determination of lactate and glucose in rat brains, which has been already described in paragraph 2.1. The GOx-PtNP-NPG/CFM and LOx-PtNP-NPG/CFM sensors, operating at 0.36 V vs. Ag/AgCl, showed a linear response to glucose and lactate, respectively, in the range 0–1 mM, following the equation Δ*I*(nA) = 2.2 × 10^−2^ + 0.78 × 10^−2^(±0.03 × 10^−2^) [glucose] (μM) and Δ*I*(nA) = 9.2 × 10^−2^ + 0.78 × 10^−2^(±0.02 × 10^−2^) [lactate] (μM). The LoD values were found to be 13 ± 2 and 6 ± 1 μM, respectively. The microbiosensors were first applied for the simultaneous determination of lactate and glucose in blood serum samples. Moreover, the basal extracellular concentrations of lactate and glucose were measured in vivo in four different rat brain structures, demonstrating the potential as a tool to investigate brain neurochemicals in vivo [37].

The number of microneedles-equipped electrochemical sensors for CGM in interstitial fluid is rapidly growing due to their high possible impact on the life quality of diabetic patients. Parrilla et al. have combined PB and Fe-Ni hexacyanoferrate in a redox mediator bilayer deposited on SPE to increase the stability of the readout, coupled with a hollow microneedle array as a sampling unit for the uptake of the interstitial fluid. The final sensing patch was optimized to achieve the desired long-term stability in the physiological range of glucose in ISF, and was employed during in vitro and ex vivo (porcine skin) tests [113].

A low-cost, swellable DA-HA hydrogel microneedle (HMN)-CGM assay has been recently reported for non-enzymatic electrochemical sensing in ISF. Pt and Ag NPs were incorporated within the hydrogel to provide electrocatalytic sites, while PEDOT:PSS was dispersed within the aqueous gel environment to increase the electrical conductivity of the patch. The sensor was tested using a type 1 diabetic rat model in hyperglycemic, euglycemic and hypoglycemic conditions. The obtained results were in agreement with the handheld glucose meter [114].

Alternative to the conventional microneedle approach, a touch-actuated biosensor was proposed including a solid microneedle array for skin penetration and a reverse iontophoresis unit for ISF extraction through the MA-created microchannels, thus increasing the glucose extraction flux by ~1.6 times when compared to RI only. The biosensor was created by drop-casting a mixture of GOx and BSA on the Au working electrode and the surface was then covered with an agarose hydrogel. The final device showed a linear response in the range 3–13 mM and a LoD of 0.92 mM and was tested during in vivo experiments using healthy and diabetic rats. The results revealed a high correlation with those obtain with a commercially blood glucometer [115].

A very interesting work by Zhang and co-workers described a new glucose-responsive microneedle (MN) array patch for self-regulated insulin delivery, employing H_2_O_2_ and pH cascade-responsive nano-sized complex micelles (NCMs). Insulin was entrapped in degradable micelles, whereas GOx in not degradable micelles, obtaining, respectively, Ins-NCMs and GOx-NCMs (Figure 7a). Under a hyperglycemic condition, the Ins-NCMs could respond to H_2_O_2_ and gluconic acid generated by the GOx-catalyzed oxidation of glucose and be dissociated in order to promote the insulin release because of the disruption of micelle structure (Figure 7a,b). The MN surface was covered by catalase nanogel (CAT-NG) embedded into the crosslinked-PVA sheath structure (Figure 7c) to reduce the skin inflammation caused by hydrogen peroxide. The investigation, performing in vivo experiments, demonstrated that the proposed patch could rapidly and safely release insulin, triggered by locally generated H_2_O_2_ and an acidic microenvironment under a hyperglycemic condition [116].

Another interesting field of application of glucose sensors is the monitoring of cell proliferation rate, since they utilize glucose as a substrate for their metabolic activities.

Madhurantakam and co-workers developed a glucose amperometric sensor to record the time-dependent glucose utilization of MiaPaCa-2 pancreatic cancer cells up to a period of 12 h. The device, operating at −0.45 V vs. Ag/AgCl, was based on CNTs and G hybrid interface coupled with GOx, and showed a wide range of detection up to glucose 21 mM, and a fast response time (<4 s) together with a high sensitivity of 0.439 μA mM^−1^ cm^−2^ [117].

## 5. Conclusions and Final Remarks

Glucose sensors represent a milestone in the development of devices for continuously monitoring biomarkers in biological fluids and diabetes management and can be considered as the first paradigm of the modern personalized medicine. Nowadays, the relevant technology is a commercial gold standard for devices that operate in both blood and interstitial fluid. On the other hand, further development and optimization steps are necessary in order to obtain even less invasive devices that can operate in sweat, tears or wound exudate, without pricking a finger to take the blood sample or using implanted electrodes. Some issues must be addressed to reach the full technological maturity and to design an appropriate sampling apparatus, a proper signal reading system and a power supply. The study of new materials plays a key role in the development of sensors compliant with the specific needs of the most advanced applications (for example by developing flexible and deformable devices capable of conforming to skin or eyes) and characterized by the high selectivity and ease of use that must exhibit the medical devices operating at the point of care. In this field, nanomaterials are attracting increasing interest because of their peculiar characteristics such as high surface area, enhanced exposure to active sites, easy functionalization, and electron transfer properties, particularly when they are metal- and carbon-based. The studies concerning sensors operating in blood represent the ideal bench test to investigate sensitivity, selectivity and the transduction mechanism of new active nanomaterials employed in devices based both on an enzymatic approach, a direct electron transfer and/or electrocatalytic processes. Nanomaterials, especially noble metals NPs or nanoporous materials, and carbon nanotubes or graphene provide a suitable environment that can be adopted for a better immobilization of enzymes, resulting in increased electron transfer and biosensing characteristics. When the presence of nanomaterials allow for DET occurrence, the literature very recently proposed the development of self-powdered devices based on two enzymatic electrodes.

Non-enzymatic sensors have been drawing great interest as a fascinating alternative to overcome the innate limitations of enzyme sensors, related to the high cost and stability issues as well as the complex and irreproducible process for mass production. As a consequence, an increasing number of publications are coming out every year describing low-cost glucose oxidation catalysts based on non-precious transition metals or their compounds, very often in mixture with carbon nanomaterials to increase electrical conductivity and/or electrocatalytical efficiency. These kind of materials have a number of advantages over other catalysts as they are available in numerous nanostructures and morphologies, are resistant to interferents such as chlorides, display high sensitivity and low detection limits but usually work well in alkaline conditions, so requiring dilution of the biological samples. In general, the biggest drawback of non-enzymatic glucose sensors is the low selectivity towards glucose. Ion selective membranes, such as Nafion or other permselective polymeric membranes, can improve selectivity since they prevent reactive molecules from reaching the electrode surface. Since interferents such as ascorbic and uric acids, dopamine and paracetamol are oxidized much faster than glucose, even lower concentrations of such interfering reactants can generate currents as high as glucose does.

When reading the recent literature as reported in this review, the main observation is that many papers claim selectivity in addition to sensitivity for glucose detection. In our opinion, the most remarkable outcome in health applications of enzyme-free systems is related to the substantial progress in selectivity gained with the nanoporous electrodes. Most nanostructured electrodes possess a very high electrochemically active surface area which generates large currents, so increasing the sensitivity of the device. This, however, does not necessarily lead to better selectivity, unless the catalytic active centers and/or the enlarged surface can amplify the glucose signals selectively. That means that, in the former case, the nanomaterial is able to decrease the overvoltage for glucose oxidation. In the latter case, if the system works in diffusion-controlled conditions (at sufficiently high potentials), all the oxidizable interferents are oxidized as soon they come in contact with the outermost surface of the nanoporous electrode. In such a way, they are depleted very rapidly inside the nanoporous structure. Considering that the glucose concentration is higher than any other electroactive molecule in blood, diffusion-controlled currents of both glucose and interfering compounds mean a predominant contribution of glucose to the signal output, and consequently, an increased selectivity for glucose measurement. Despite the research innovations related to the synthesis of nanoporous materials, in terms of commercialization, non-enzymatic electrochemical sensors have yet a long way to go.

Furthermore, the recent literature is proposing new captivating challenges that could open up novel lines of research. The most ambitious topics are:(i)The development of multi-sensor platforms capable of detecting other biological parameters together with glucose concentration, in order to provide a deeper clinical picture of the patient’s physiological state;(ii)The development of a non-invasive signal readout system, which does not require complex electronics and/or an external power supply. Embedding electronic component into everyday objects is very laborious and leads to an increase in costs that is not compatible with mass production. The development of self-powered sensors or sensors based on passive radio frequency identification reading represents an elegant solution to solve this problem. Signal amplification through a transistor architecture can also help us cope with this technological challenge;(iii)The use of a sampling system allows the collection of the biofluid of interest in a continuous way, in order to obtain truly real-time data that are not influenced by the stagnation of the sample near the transducer. The fabrication of microfluidic systems in PDMS or other medical grade absorbent materials are interesting solutions to meet this need. Reverse iontophoresis for the extraction of interstitial fluid from the skin, thanks to the application of a potential, can be a valid tool to integrate into these devices;(iv)The design of tattoo sensors that can be applied directly to the patient’s skin without the use of a support substrate has the potential to bring the concept of wearable electronics to a new stage, with the aim to improve the patient comfort, as well as the economic impact and fingerprint of the monitoring device;(v)The continuous and tight glycemic monitoring and control by combining highly efficient sensing units with closed-loop systems to obtain optimal and timely drug administration.

We firmly believe that the research on new nanomaterials can make a significant contribution in the development of these new research topics thanks to their fascinating and unique properties.

## Figures and Tables

**Figure 1 nanomaterials-13-01883-f001:**
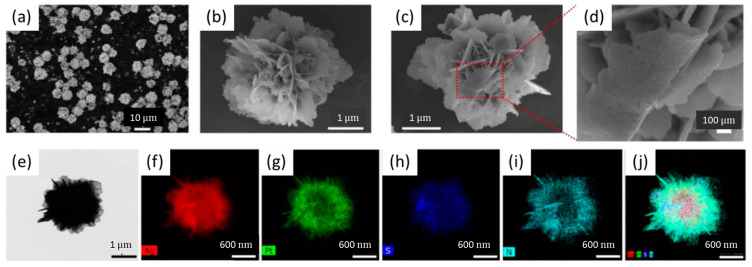
(**a**) SEM image of AgNFs-Pt@BSA nanoparticles. FESEM images of (**b**) AgNFs@BSA and (**c**) AgNFs-Pt@BSA nanoparticles. (**d**) Enlarged FESEM image of a petal from the red dotted box in (**c**). (**e**) FETEM bright-field image of AgNFs-Pt@BSA. FETEM EDS maps of AgNFs-Pt@BSA nanoparticles in various colors: (**f**) red dots represented Ag atoms, while (**g**) green dots represented Pt atoms; (**h**) navy dots represented S atoms, and (**i**) cyan dots represented N atoms. (**j**) A superposition FETEM EDS map of AgNFs-Pt@BSA nanoparticles by Ag, Pt, S and N atoms. © IOP Publishing. Reproduced with permission from [34]. All rights reserved.

**Figure 2 nanomaterials-13-01883-f002:**
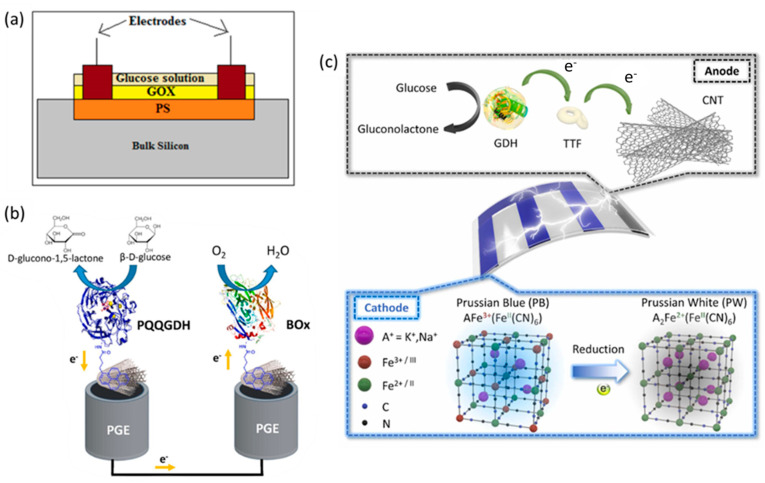
Enzymatic blood glucose sensors. (**a**) Schematic diagram of experimental setup for I-V (current–voltage) measurement. © IOP Publishing. Reproduced with permission from [39]. All rights reserved. (**b**) Scheme of the self-powered biosensor based on PGE transducers and PBSE immobilization of PQQ-dependent glucose dehydrogenase (PQQGDH) on the anode and Bilirubin oxidase (BOx) on the cathode. Reprinted with permission from [40]. Copyright 2021 Elsevier. (**c**) Illustrations of the redox reaction mechanism of glucose biosensor with sequence of color change of PB bars (reduction reaction of PB bars) that occurred by the transfer of electrons produced by GOR of anode. Reprinted with permission from [41]. Copyright 2022 Elsevier.

**Figure 4 nanomaterials-13-01883-f004:**
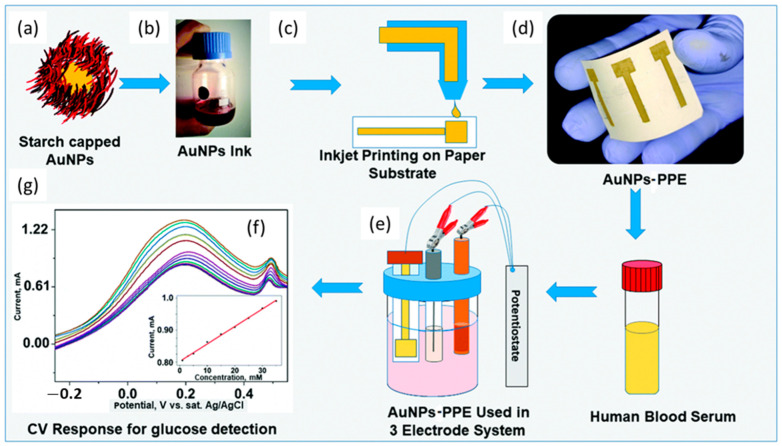
(**a**) synthesis of starch capped AuNPs, (**b**) aqueous solution of Au-nano-ink, (**c**) inkjet-printing of the Au-ink, (**d**) AuNP-PPE, (**e**) human blood serum testing, (**f**) AuNP-PPE used as a working electrode and (**g**) CV responses of glucose in CV. Reproduced from [54] with permission from the Centre National de la Recherche Scientifique (CNRS) and the Royal Society of Chemistry.

**Figure 5 nanomaterials-13-01883-f005:**
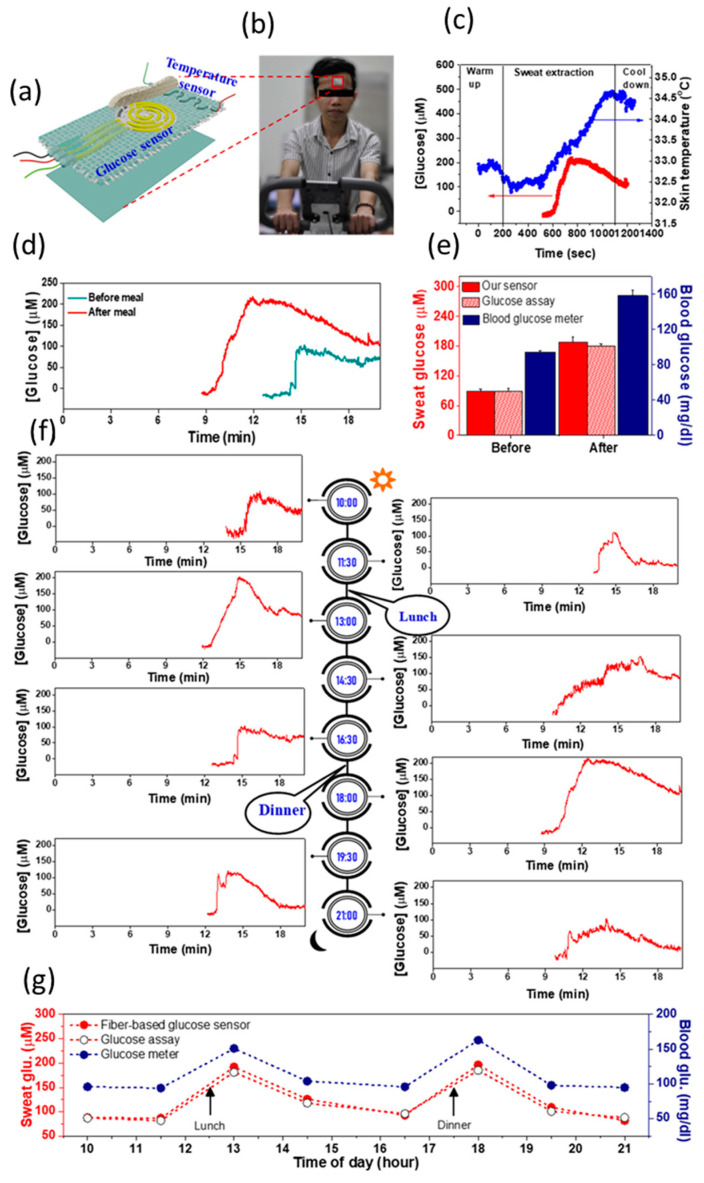
WSNF-based glucose sensor patch integrated with stretchable fabric for continuous on-body monitoring of glucose levels in human sweat. (**a**) Schematic diagram of the WSNF glucose sensor on stretchable fabric integrated with a stretchable temperature sensor. (**b**) The fabricated device was attached to the forehead of a subject for (**c**) continuous monitoring of skin temperature and the glucose levels in sweat. (**d**) Continuous on-body monitoring of glucose levels in sweat before and after meals. (**e**) Comparison of the glucose levels in sweat detected by the WSNF-based stretchable glucose sensor patch, a commercial sweat glucose assay, and the blood glucose level measured by a blood glucose meter. (**f**) Continuous on-body monitoring of glucose levels in sweat for 1 day. (**g**) Summary of the glucose levels measured in 1 day by the WSNF sweat glucose sensor, the commercial sweat glucose assay and a blood glucose meter. Reprinted with permission from [82]. Copyright 2019 American Chemical Society.

**Figure 6 nanomaterials-13-01883-f006:**
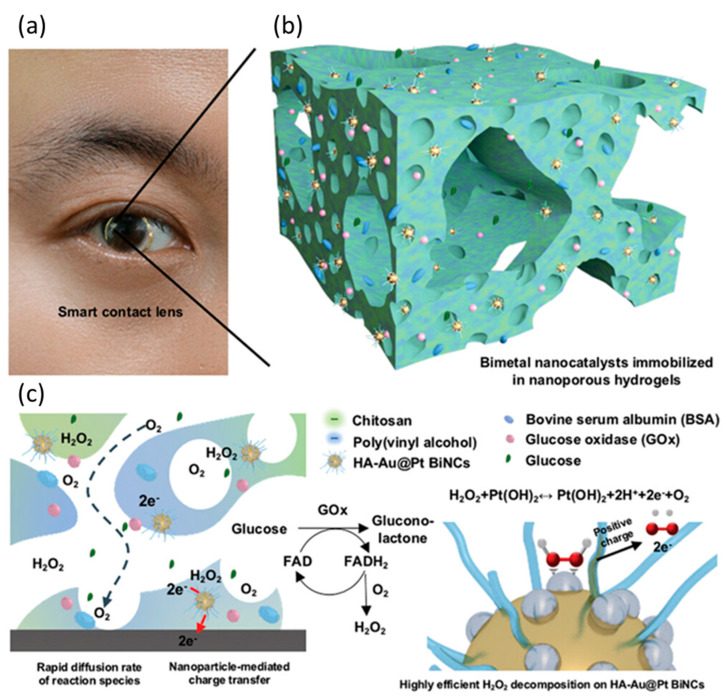
(**a**) Schematic illustration of smart contact lens for diabetes monitoring. (**b**,**c**) The structure and glucose sensing mechanism of bimetallic nanocatalysts (BiNCs) in nanoporous hydrogels. Flavin adenine dinucleotide (FAD) inside glucose oxidase in the hydrogels undergoes a redox reaction with diffused glucose and O_2_, which is then reduced to FADH_2_. H_2_O_2_ are rapidly decomposed on the surface of HA-Au@Pt BiNCs and generate two electrons which are rapidly transported to the electrode surface by nanoparticle-mediated charge transfer. The nanopores in the hydrogels play important roles for the rapid diffusion of reaction species and rapid swelling with superabsorbent properties. Reprinted with permission from [108]. Copyright 2022 WILEY–VCH.

**Figure 7 nanomaterials-13-01883-f007:**
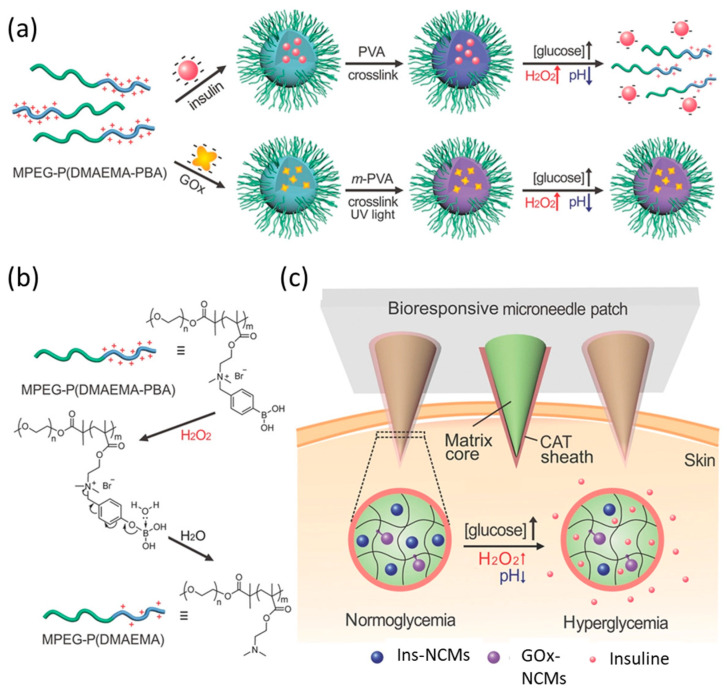
Schematic of the glucose-responsive insulin delivery system utilizing H_2_O_2_ and pH cascade-responsive NC-loading MN-array patch (abbreviation: MPEG polyethylene glycolyl monomethyl ether 2-bromoisobutyrate, DMAEMA 2-(dimethylamino)ethyl methacrylate, PBA 4-(bromomethyl)phenylboronic acid, CAT catalase). (**a**) Formation of Ins-NCMs and GOx-NCMs and mechanism of glucose-responsive insulin release. (**b**) Schematic of H_2_O_2_-triggered charge reduction of the polymer. (**c**) Schematic of the NCM-containing MN-array patch with a CAT sheath structure for in vivo insulin delivery. Insulin release is triggered under a hyperglycemic state. Reprinted with permission from [116]. Copyright 2018 WILEY–VCH.

**Table 1 nanomaterials-13-01883-t001:** Main analytical features of sensors operating in serum or blood.

Electrode Modifier	Enzyme/Active Material	Sensitivity (μA mM^−1^ cm^−2^)	LoD (μM)	Response Time	Ref.
GDH/CS/MWCNT	GDH	-	15.6	-	[28]
GOx/G/PtNPs/NF	GOx	7.87	-	-	[33]
GOx/GA/AgNFs-Pt@BSA	GOx	305	300	-	[34]
GOx/Pt NPs/nPG	GOx	254	13		[37]
GDH/nPG	GDH	-	15	-	[38]
GOx/nanoporous Si	GOx	-	-	170 ms	[39]
PQQGDH/PBSE/MWCNT	GDH	77.7	4	-	[40]
GDH/TTF/CNT	GDH	16.4	15	-	[41]
CuO/nanoporous Cu	CuO	1621	0.2	3	[43]
CuO nanowire/nanoporous Cu	CuO	1925	1	1.5	[44]
CeO_2_@CuO	CeO_2_@CuO	3320	0.019	>1	[46]
CuxO-NiO NC	CuxO-NiO	1380	7.3	-	[47]
LIG/Cu_2_O and Au	CuO	236	0.31	<15	[49]
G/Co(OH)_2_ NPs	Co(OH)	2410	0.67	<3	[50]
*β*-Ni(OH)_2_/Ce_2_O_3_/rGO	*β*-Ni(OH)_2_/Ce_2_O_3_	33.1	13	-	[51]
micronano dualporous Au	Au	48.4	25	2.5	[53]
AuNPs/PPE	AuNPs	-	10	-	[54]
Nanoporous Pt	Pt	358	-	<5	[56]
CuCe NPs/G or CNT	CuCe NPs	-	0.095	-	[57]
CuNi nanoalloy	CuNi	18.2	0.02	5 s	[52]
Ni_3_N-PCF/Nafion	Ni_3_N	1630	0.05	1.7	[58]
Ni_x_Co_3–x_N/NG	Ni_x_Co_3–x_N	1800	0.05	<3 s	[59]
7,7,8,8-tetracyanoquinodimethane	7,7,8,8-tetracyanoquinodimethane	29,670	0.01	3 s	[60]
Co phosphide/nanoporous Cu	Co phosphide/nanoporous Cu	1818	0.4	4	[42]
Co_3_(BTC)_2_ MOF	Co_3_(BTC)_2_	1792	0.33	2.25	[64]
CuOx@Co_3_O_4_/Co-ZIF-67 compound	CuO_x_@Co_3_O_4_	27,800	0.036	1	[65]
NiCo/C	NiCo	265	0.2	-	[62]
Cu NPs/ZIF-8	Cu NPs	1590	2	-	[66]

**Table 2 nanomaterials-13-01883-t002:** Typical glucose concentrations for healthy and diabetic people in physiological fluids.

Physiological Fluid	Glucose Concentration for Healthy Patients/mM	Glucose Concentration for Diabetic Patients/mM
Blood	4.9–6.9	2–40
Interstitial fluid	3.9–6.6	1.99–22.2
Urine	2.78–5.55	>5.55
Sweat	0.06–0.11	0.01–1
Saliva	0.23–0.38	0.55–1.77
Tears	0.05–0.5	0.5–5

**Table 3 nanomaterials-13-01883-t003:** Analytical features of all the glucose sensors described for urine analysis.

Working Electrode	Active Material	Sensitivity (μA mM^−1^ cm^−2^)	LoD (μM)	Response Time (s)	Ref.
Co_3_O_4_/PPy core–shell nano-heterostructures/Ni-foam	Co_3_O_4_/PPy	1594	740	12	[66]
Co(OH)_2_ NPs/3D G	Co(OH)_2_ NPs	2410	0.67	3 s	[50]
Cu_x_O-NiO NC	Cu_x_O-NiO	1380	7.3	-	[47]
Cu/CuO/Cu(OH)_2_/polydopamine	Cu/Cu(II) nano-heterostructures	223	20	<3	[72]
MWCNTs modified with N salicylaldehyde, N′-2hydroxyacetophenon-1,2phenylene diimino Ni(II) complex	N salicylaldehyde, N′-2hydroxyacetophenon-1,2phenylene diimino Ni(II) complex	-	1300	-	[73]
NiCo-S nanosheets/Carbon nanohorns	NiCo-S nanosheets	1840	5.6	1.7	[74]
Fe_2_O_3_ NPs	Fe_2_O_3_ NPs	-	0.044	-	[75]
ZnO nanorod/Ru–C_3_N_4_	ZnO nanorod/Ru–C_3_N_4_	346	3.5	3	[76]
ZrO_2_-Ag-G-SiO_2_	ZrO_2_-Ag-G	-	-	1	[77]
LIG/Au@CuO/V_2_CT_x_ MXene	Au@CuO/V_2_CT_x_ MXene	-	1.8	1.5	[78]
rGO modified with Aconitum heterophyllum plant root	Aconitum heterophyllum plant root	61.76			[80]
GOx/BSA/GA/Pt/CNT	GOx	38	15.5		[81]

**Table 4 nanomaterials-13-01883-t004:** Analytical features of all the glucose (bio)sensors operating in sweat.

Working Electrode	Enzyme/Active Material	Sensitivity (μA mM^−1^ cm^−2^)	LoD (μM)	Response Time (s)	Ref.
Au nanowrinkles/nanohybrid fibers rGO/PU reduced graphene oxide–polyurethane	Au	140	0.5	-	[82]
PB/CS-GOx-Au NPs	GOx	1.27	74	-	[83]
Au NPs/Tannic acid-3-aminopropyltriethoxysilane/aCC	Au NPs	72	3.3	4	[84]
GOx/PtNPs/hydroxyethyl cellulose/3D laser-scribed graphene	GOx	69.64	0.23	-	[85]
GOx/CS/G/PB	GOx	1.790	2.45		[86]
GOx/PANI/PVA	GOx	-	0.2	-	[87]
GOx/rGO-Au	GOx	53.7	-	-	[88]
GOx/CS/PB_PEDOT	GOx	-	-	-	[89]
GOx-BSA-GA/Nanoimprinting Au	GOx	1.76	55	-	[90]
ZnO tetrapods and MXene nanoflakes	ZnO tetrapods and MXene nanoflakes	29.88	21	-	[91]
CeAlO_3_-Carbon nitride	CeAlO_3_ and Carbon nitride	68,718	0.00086		[92]
CNT/CuO NC	CuO	-	3.9	2	[93]
GO nanosheets supporting Au NF	Au NF	474	123	6	[94]
Ni/Sm-LDH NPs/Mxene	Ni/Sm-LDH NPs	1673	0.24	-	[95]
Pt NPs/MXene	Pt NPs/MXene	3.43	29		[96]
NiCo-MOFs	NiCo-MOFs	1422	0.11		[97]
nitrogen-doped carbon nanosheets containing Co and Cu	Co-Cu	1489	0.34		[98]
2D nanosheets of (Ni and Co) MOFs		2701	0.09		[99]
Ni–Co MOF nanosheet		205	4.25		[100]
Cu NPs ink		2690	0.5	1.5	[101]
AuNPs-rGO NC		-	0.01	-	[102]
glucuronic acid-Au NFs		7.0913	5	-	[103]

## Data Availability

Not applicable.

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
