# Peer review of "Focus Review on Nanomaterial-Based Electrochemical Sensing of Glucose for Health Applications"

_nanomaterials, 2023, doi:10.3390/nano13121883_

Round 1

Reviewer 1 Report

Please consider the document in attachment.

Reviewer 2 Report

Good review. The anti-fouling properties of nanomaterials-based electrochemical glucose sensors and the shortage of the selectivity of non-enzymatic sensors should be discussed. Besides, I do not think that non-enzymatic electrochemical glucose sensing is meaningful and useful in practice due to the absence of selectivity. As a sensor, the selectivity is more important than sensitivity. 

No comment

Reviewer 3 Report

The authors aim to give a comprehensive overview of the most relevant advancements in glucose sensing achieved in the last 5 years. Some innovative electrochemical sensing strategies, based on nanomaterials, have been thoroughly discussed on their analysing performances, advantages and limitations. The authors also provide some critical reviews on the major analytical challenges in glucose detection, in blood and serum samples, as well as in less conventional biological fluids. The review is comprehensive and informative and can be published after some major issues being addressed:

1/ Tables are needed to list the glucose sensors fabricated by similar electrodes with different modifiers.

2/ The detection mechanisms for non enzymatic electrochemical sensors are highly suggested to be graphically illustrated.

3/ It is highly suggested to add the information on the electrochemical sensors empolying biomaterials such as DNA and aptamer. ( For instance: https://doi.org/10.1002/celc.202000105; https://doi.org/10.1016/j.bios.2022.114251)

4/ There are some typo errors in the text, please revise them carefully.

The authors aim to give a comprehensive overview of the most relevant advancements in glucose sensing achieved in the last 5 years. Some innovative electrochemical sensing strategies, based on nanomaterials, have been thoroughly discussed on their analysing performances, advantages and limitations. The authors also provide some critical reviews on the major analytical challenges in glucose detection, in blood and serum samples, as well as in less conventional biological fluids. The review is comprehensive and informative and can be published after some major issues being addressed:

1/ Tables are needed to list the glucose sensors fabricated by similar electrodes with different modifiers.

2/ The detection mechanisms for non enzymatic electrochemical sensors are highly suggested to be graphically illustrated.

3/ It is highly suggested to add the information on the electrochemical sensors empolying biomaterials such as DNA and aptamer. ( For instance: https://doi.org/10.1002/celc.202000105; https://doi.org/10.1016/j.bios.2022.114251)

4/ There are some typo errors in the text, please revise them carefully.

Round 2

Reviewer 1 Report

In my opinion the manuscript is ready for publication.

Reviewer 3 Report

Accept